# Assessing Trail Running Biomechanics: A Comparative Analysis of the Reliability of Stryd^TM^ and GARMIN_RP_ Wearable Devices

**DOI:** 10.3390/s24113570

**Published:** 2024-06-01

**Authors:** César Berzosa, Cristina Comeras-Chueca, Pablo Jesus Bascuas, Héctor Gutiérrez, Ana Vanessa Bataller-Cervero

**Affiliations:** 1Faculty of Health Sciences, Universidad San Jorge, Autov. A-23 km 299, 50830 Villanueva de Gállego, Spain; cberzosa@usj.es (C.B.); pbascuas@usj.es (P.J.B.); hgutierrez@usj.es (H.G.); avbataller@usj.es (A.V.B.-C.); 2ValorA Research Group, Health Sciences Faculty, Universidad San Jorge, 50830 Villanueva de Gállego, Spain

**Keywords:** wearable technology, running metrics, outdoor testing, trail running analysis

## Abstract

This study investigated biomechanical assessments in trail running, comparing two wearable devices—Stryd Power Meter and GARMIN_RP_. With the growing popularity of trail running and the complexities of varied terrains, there is a heightened interest in understanding metabolic pathways, biomechanics, and performance factors. The research aimed to assess the inter- and intra-device agreement for biomechanics under ecological conditions, focusing on power, speed, cadence, vertical oscillation, and contact time. The participants engaged in trail running sessions while wearing two Stryd and two Garmin devices. The intra-device reliability demonstrated high consistency for both GARMIN_RP_ and Stryd^TM^, with strong correlations and minimal variability. However, distinctions emerged in inter-device agreement, particularly in power and contact time uphill, and vertical oscillation downhill, suggesting potential variations between GARMIN_RP_ and Stryd^TM^ measurements for specific running metrics. The study underscores that caution should be taken in interpreting device data, highlighting the importance of measuring with the same device, considering contextual and individual factors, and acknowledging the limited research under real-world trail conditions. While the small sample size and participant variations were limitations, the strength of this study lies in conducting this investigation under ecological conditions, significantly contributing to the field of biomechanical measurements in trail running.

## 1. Introduction

Trail running, recently recognized by the International Association of Athletics Federations (IAAF) as an emerged running discipline, has rapidly increased in popularity, which has led to a growing scientific interest in the field of sports science [1]. The inherent difficulty of trail running, often characterized by positive and negative elevation changes in the terrain, increases the energy and muscular demand during its practice and heightens the importance of the strategic combination of these metabolic pathways and biomechanical conditions for optimal performance [1]. The complexity of this sport involves numerous performance factors, including aerobic capacity, anaerobic threshold, muscular strength, and resistance to force due to both positive and negative elevation changes [2,3]. Additionally, considerations such as thermoregulation, running economy, and biomechanical adaptations in uphill and downhill running play a crucial role [2,3]. These adaptations encompass changes in the foot strike pattern, joint kinematics, and energy cost [2,3]. This is why biomechanical variables such as power, speed, cadence, and contact time are worthy of evaluation due to their close relationship with performance. In athletic disciplines, power denotes the rate of performing work and is a determinant of explosive strength, which is essential for activities necessitating rapid force application. Speed is the scalar quantity representing the distance covered per unit of time, which is pivotal in disciplines requiring swift transit. Cadence, the frequency of stride or pedal revolutions per minute, is a critical factor in endurance sports, influencing the metabolic cost and endurance capacity [1,2]. Moreover, vertical oscillation and contact time are biomechanical parameters in gait analysis. Vertical oscillation quantifies the vertical displacement of the center of mass during locomotion, while contact time measures the duration of foot–ground interactions [4]. These parameters are indicative not only of performance, running economy, and biomechanical efficiency, but also of injury risk [5].

However, biomechanical measurements under ecological conditions according to the terrain typology in trail athletes are uncommon and limited to laboratory exercise tests [1,6]. The past decade witnessed significant technological advancements in wearable sports devices for assessing running biomechanics. These devices enable the measurement of training loads and the development of training protocols [7]. This biomechanical evaluation of running is usually performed in a laboratory using treadmills, force plates, and motion capture systems. Nevertheless, this approach is usually both inaccessible and too expensive for practitioners, emphasizing the need for more affordable methods suitable for outdoor use.

The significance of monitoring biomechanical parameters measured under real-world conditions during trail running has increased owing to their association with performance. It is necessary to objectively assess biomechanical parameters under ecological conditions to quantify the training load and performance [6]. The availability and popularity of wearable sports technology for running has grown extensively in recent years [8]. In this context, recent advances in wearable technology have accelerated the development of less obtrusive and more precise and affordable devices designed to monitor a wide range of parameters in trail runners exercising in their normal environments [8]. Recently, there has been an effort to produce low-cost, portable gait and running analysis equipment. This has allowed researchers to remove participants from an artificial laboratory environment and measure participants in a more natural environment [9]

Commercially available wearable technology, and the biofeedback provide by them, has been welcomed by coaches and runners, including those in the trail running community. The commercial availability of such devices in the market is extensive, including popular brands such as GARMIN_RP_, Stryd^TM^, RunScribe, Polar, and Suunto, among others. These devices, equipped with Global Positioning System (GPS), Inertial Measurement Unit (IMU) sensors, and more, provide insights into running metrics. These devices are being used to quantify the training load by reporting data such as cadence, stride length, power, contact time, or vertical oscillation [10]. However, more high-quality research is needed to determine the accuracy and reliability of measurements obtained by this new technology, which is continuously developing and advancing [11], but is still in the exploratory phase [12].

In the market, there are two standout wearable devices designed for measuring running biomechanics on the market. On one hand, there is the Stryd Power Meter, which is a foot-pod device that has been used recently to measure variables related to trail running performance [13] and this device is capable of measuring power, contact time, flight time, step length, vertical oscillation, and cadence [6,7,9]. On the other hand, there are devices from the GARMIN_RP_ brand. Apart from measuring heart rate and estimating energy expenditure, these devices provide data on distance, speed, and elevation [14], vertical oscillation [15], contact time [16], and cadence [8]. While these kinds of devices have been extensively studied in laboratory settings using treadmills for testing, there is a notable dearth of research in real-world scenarios, conducting tests under natural conditions. This gap in knowledge is particularly significant in the context of trail running, where the complexity arises from the diverse terrain characteristics [6]. The comparison between motorized treadmill running and overground running highlights variations in sagittal plane measures and spatiotemporal parameters, with conflicting findings across analyses of kinematics, kinetics, muscle activity, and muscle-tendon outcomes [17]. Notably, these comparisons do not exclusively involve trail running, which presents unique characteristics in overground locomotion [17]. Thus, it is imperative to understand how these devices measure in real environmental and terrain conditions, specifically whether these devices demonstrate high intra-device reliability and exhibit agreement between device measurements, which is essential to determine the interchangeability of these devices in practical applications. Moreover, investigating these aspects on both uphill and downhill terrains provides insights into potential variations, ensuring a comprehensive evaluation of device performance across diverse slope conditions. This gap can be addressed by focusing on assessing how the Stryd^TM^ and GARMIN_RP_ devices perform in real-world conditions, specifically exploring their intra-device reliability and agreement between measurements.

This study aimed to investigate and compare the biomechanical data reported by the two most popular wearable devices, GARMIN_RP_ and Stryd^TM^, which can provide measures of biomechanics during trail running under natural conditions. Specifically, our objectives include, on the one hand, assessing the consistency and agreement of biomechanical measurements within the GARMIN_RP_ or Stryd^TM^ devices by comparing one device against the other. On the other hand, we sought to investigate the inter-device reliability and agreement by comparing the biomechanical parameters measured by a Stryd^TM^ device against a GARMIN_RP_ device. By conducting these comparisons, we aimed to provide insights into the reliability and consistency of these wearable devices in measuring key biomechanical variables, offering valuable information for both researchers and practitioners in the field of trail running and sports science.

## 2. Materials and Methods

### 2.1. Participants

The five participants (four males and one female) included in the study were healthy young adults with an average age of 33 ± 4.4 years. The participants were required to have a minimum of one year of experience in trail running. The average body mass index (BMI) of the participants was 22.6 ± 2.2 kg/m^2^. Before testing, the participants were informed about the procedure and study protocol, and signed informed consent was collected from each participant. This study was performed in accordance with the ethical guidelines of the Helsinki Declaration of 1964 (revised in Fortaleza, 2013) [18]. The study was reviewed and approved by the Ethics Committee of the University of San Jorge (code No. 005–19/20).

### 2.2. Study Design

To assess the inter-device reliability and intra-device reliability of the GARMIN_RP_ and Stryd^TM^ wearable devices, we compared the recorded data from both devices on a field running track and the data collected from both sessions were pooled together. The variables analyzed were power, speed, cadence, vertical oscillation, and contact time for both devices. While there are other wearable devices, such as Runscribe, the decision to include Stryd was based on the necessity for the runner to wear all devices. To minimize potential interference, particularly related to device positioning, Stryd was chosen, as it demonstrated higher reliability in measuring power compared to Runscribe [19], and has the closest agreement with the theoretical power output [20]. Additionally, the findings from Kozinc et al. [21] revealed an unacceptable coefficient of variation for power, foot strike type, and horizontal ground reaction force rate.

The participants completed two sessions of a trail running course, with a 2.5 km distance and a 195 m elevation gain, followed by a descent along the same route, as illustrated in Figure 1. The tests were conducted at the same time of day on both days. A one-week recovery period was implemented between sessions for the participants.

### 2.3. Equipment

On the left wrist of the participants, two Garmin_RP_ Fenix 7S Solar watches (Garmin Ltd., Southampton, UK) were placed. This device is capable of detecting running biomechanics, activity, and sleep using several sensors, including a triaxial accelerometer, a Global Navigation Satellite System (GNSS) sensor, including GPS functionality, and a photodiode sensor for photoplethysmography measurements. This device measures running biomechanics variables such as power, speed, cadence, vertical oscillation, and ground contact time. The participants also wore a GARMIN_RP_ HRM-PRO heart rate monitor below the pectoral zone in a centered and vertically oriented position. Power was assessed by the GARMIN_RP_ HRM-PRO heart rate monitor and speed, cadence, vertical oscillation, and ground contact time were measured by the GARMIN_RP_ Fenix 7S Solar watch. The measured power specifically refers to the external mechanical power, calculated from force (estimated via accelerations using accelerometers) and velocity. This includes the work exerted by runners during both the loading phase and subsequent push-off to counteract environmental factors such as ground reaction force, gravity, and surface friction.

On the right foot of the participants, two Stryd^TM^ foot pods (Stryd Powermeter; Stryd, Inc., Boulder, CO, USA) were attached. This device is a carbon fiber-reinforced foot pod based on a 6-axis inertial motion sensor (3-axis gyroscope and 3-axis accelerometer). The variables measured with this Stryd^TM^ technology include power, speed, cadence, vertical oscillation, and the duration of contact time.

Figure 2 illustrates the equipment that the runners wore during the trial. The footwear during the test was chosen by each participant, according to their personal preferences. The runners also carried two Android smartphones with the Stryd^TM^ application in a waist pack to collect data from the devices. At the beginning of the test, the weight, height, and age of the participants were entered into both the watches and the Stryd mobile application. Two researchers simultaneously initiated the watches and mobile devices. Subjects were asked to stop both the watches and the mobile app upon completing the descent.

### 2.4. Data Extraction

The data capture frequency was 1 Hz for both devices (Stryd^TM^ and GARMIN_RP_), a sampling frequency similar to that used in similar studies on running biomechanics [7,8,16]. Following data recording, the data were exported for subsequent analyses. The data upload to the Garmin Connect software, version 5.1, on a PC was conducted to ensure synchronization with GPS time stamps, resulting in the acquisition of files in “.fit” format. Conversely, the Stryd^TM^ device employs a distinct approach, utilizing the smartphone for data upload and relying on GPS time stamps for temporal synchronization. After data transmission to the Stryd^TM^ platform, the files were downloaded in “.fit” format. Both sets of files were then processed through the free license for GoldenCheetah software, version 3.6, which facilitated the temporal synchronization of all files possessing timestamp marks (.fit) before collectively exporting them to Excel(Version 2110). After synchronization, both files had the same number of samples for analysis.

### 2.5. Statistical Analysis

The Statistical Package for the Social Sciences (SPSS) version 23.0 (SPSS Inc., Chicago, IL, USA) was used to perform all the statistical analyses. Statistical significance was set at *p* < 0.05 for all tests. The data are presented as the mean ± standard deviation (SD).

Measurements obtained from both sessions were pooled together. To determine the intra-device reliability, the measurements, including power, speed, cadence, vertical oscillation, and contact time, obtained from one device (either GARMIN_RP_ or Stryd^TM^) were compared to those recorded by the second device of the same brand. Moreover, the measurements of the running biomechanics obtained from the Stryd^TM^ devices were compared to those recorded by the GARMIN_RP_ devices to determine the inter-device reliability.

To evaluate agreement in inter- and intra-class measurements, the Intraclass Correlation Coefficient (ICC) and its 95% confidence limits were utilized. Values less than 0.5 are indicative of poor reliability, values between 0.5 and 0.75 indicate moderate reliability, values between 0.75 and 0.9 indicate good reliability, and values greater than 0.90 indicate excellent reliability [22]. Additionally, the Coefficient of Variation (CV, %) was calculated to assess the relative variability of the measurements. For intra-device comparisons, measurements from one GARMIN_RP_ device were compared to a second GARMIN_RP_ device, both worn by the same individual on the same wrist. Similarly, measurements from one Stryd^TM^ device were compared to a second Stryd^TM^ device, both worn by the same individual on their foot. As stated before, the tests were conducted on different days following the comparison protocol.

In order to assess the strength of the linear relationship between variables, the Pearson correlation coefficient was employed as a measure in this study [23]. Furthermore, scatter plots were created to visually examine the relationship between the measurements from two devices. Simple linear regression analysis was employed to quantify these potential relationships. The Bland–Altman method was utilized to assess the distribution of residuals from the regression analysis [24]. These analytical approaches collectively aimed to provide insights into the intra- and inter-device reliability and consistency of the wearable devices.

## 3. Results

The study included a total of five participants who completed two separate sessions of a trail running course and the trial recording was programmed to capture one sample per second, resulting in a total of 19,731 samples for GARMIN_RP_ and 17,917 records for Stryd^TM^. After synchronization, the number of samples became identical. Table 1 displays the descriptive data for each device, presenting the mean value and standard deviation of all samples for the entire course and categorized by positive and negative slopes.

### 3.1. Intra-Device Reliability Analysis

The reliability analysis provided important findings regarding the consistency of the GARMIN_RP_ and Stryd^TM^ measurements in assessing running metrics. The ICC consistently demonstrated a high reliability level, exceeding 0.90 (from 0.939 to 0.997) for power, speed, cadence, vertical oscillation, and contact time in both the GARMIN_RP_ and Stryd^TM^ devices. These results provide insights into the capacity of these devices to provide consistent measurements in a trail running context.

The CV further confirmed the reliability, indicating low measurement variability. Across all variables, including a separate analysis by positive and negative slope, the calculated CV values ranged from 0.3% to 3.3% for GARMIN_RP_ and from 0.4% to 2.4% for Stryd^TM^, attesting to a low level of measurement variability. Furthermore, an in-depth examination focusing on the differentiation between positive and negative slopes, demonstrated a consistent CV range of 0.31% to 2.4% for GARMIN_RP_ and 0.4% to 2.8% for Stryd^TM^, indicative of a prevailing trend of minimal variability in the measured variables.

A comprehensive Bland–Altman analysis showed narrow limits of agreement (LoA) for both GARMIN_RP_ and Stryd^TM^, indicating minimal systematic bias between the measurements obtained from one device compared to the other for power, speed, cadence, vertical oscillation, and ground contact time. Additionally, strong positive correlations were observed between the measurements of both devices for both GARMIN_RP_ and Stryd^TM^, with correlation values ranging from 0.845 to 0.993.

The linear regression analyses for the running metrics revealed statistically significant results across all variables (*p* < 0.01). The obtained B-values and t-values consistently showed high values, indicating robust associations between the variables obtained by both devices for both GARMIN_RP_ and Stryd^TM^. Importantly, the narrow confidence intervals underscored the precision of the estimates. These findings support a strong relationship between the GARMIN_RP_ devices (GARMIN_RP_ 1 vs. GARMIN_RP_ 2) and Stryd^TM^ devices (Stryd^TM^ 1 vs. Stryd^TM^ 2), emphasizing their reliability and consistency in measuring diverse performance metrics. Table 2 provides a comprehensive overview of all results regarding the reliability of devices, differentiated by GARMIN_RP_ and Stryd^TM^.

### 3.2. Inter-Device Agreement

The reliability analysis revealed notable distinctions in the inter-device reliability when comparing the GARMIN_RP_ to Stryd^TM^ measurements for running metrics. The ICC consistently indicated a high level of reliability for speed, cadence, and contact time (ranging from 0.841 to 0.991), and a lower level of reliability for uphill power and contact time and downhill vertical oscillation.

Similarly, the calculated CV values pointed towards increased measurement variability when comparing GARMIN_RP_ to Stryd^TM^ devices. The CV indicated a high variability for power for uphill speed, cadence, and contact time and for downhill vertical oscillation.

The Bland–Altman analysis displayed wider limits of agreement (LoA), suggesting a higher degree of systematic bias between the measurements obtained from GARMIN_RP_ compared to Stryd^TM^. Figure 3 illustrates the Bland–Altman plots comparing GARMIN_RP_ vs. Stryd^TM^ for power, speed, vertical oscillation, and ground contact time on the uphill and downhill gradients. All the Bland-Altman analysis graphs, both for the entire course of the test and divided into uphill and downhill segments, have been included in the Appendix A. However, the Pearson correlation analysis demonstrated a high correlation between the measurements of the GARMIN_RP_ and Stryd^TM^ devices, highlighting a strong relationship despite the observed differences in inter-device reliability.

Univariate Linear Regression analyses showed statistically significant results (*p* < 0.01) but with lower B-values and t-values, indicating weaker associations between the variables obtained from different devices. This weaker association was particularly evident in the power and speed variables and can be seen in Figure 4. Table 3 provides a comprehensive overview of all the results regarding the reliability of the devices, comparing GARMIN_RP_ and Stryd^TM^. The resulting graphs from the Univariate Linear Regression analyses, for both the entire course of the test and segmented into uphill and downhill sections, have been included in the Appendix A.

## 4. Discussion

The major finding was that, while both devices showed high intra-device reliability, distinctions emerged in the inter-device reliability between the GARMIN_RP_ and Stryd^TM^ measurements for certain running metrics. According to the reliability analysis, both GARMIN_RP_ and Stryd^TM^ devices exhibited high intra-device reliability, with the ICC consistently surpassing 0.90, and the CV confirmed low measurement variability across all variables and slopes. The Bland–Altman analysis revealed narrow limits of agreement, indicating minimal systematic bias and strong positive correlations between measurements from both devices, and the linear regression analyses demonstrated statistically significant results across all variables. On the other hand, concerning the inter-device reliability analysis, the ICC indicated a high level of reliability for speed, cadence, and contact time, but a lower level for uphill power and contact time and downhill vertical oscillation. The CV values pointed to increased measurement variability when comparing GARMIN_RP_ to Stryd^TM^ devices. The Bland–Altman analysis displayed wider limits of agreement, suggesting a higher degree of systematic bias between the measurements obtained, despite a strong overall correlation. Lastly, the Univariate Linear Regression analyses showed statistically significant results, but with lower B-values and t-values, indicating weaker associations between the variables obtained from the different devices, which were particularly evident in the power and speed variables.

In light of intra- and inter-device reliability considerations, our findings are consistent with the results reported in previous scientific studies [6,7,8,9,13,14,15,16]. There is a large amount of evidence about determining validity through comparing these devices to gold standards, but there is limited research on intra-device reliability. Furthermore, most studies were conducted in laboratory settings using treadmills, with very few articles investigating the measurement quality of the assessments of these wearable devices in natural field conditions.

### 4.1. Stryd^TM^ and GARMIN_RP_ vs. Gold Standard

Some discrepancies were detected when comparing the results of running biomechanics measured by wearable devices such as Stryd^TM^ and GARMIN_RP_ with those of the gold standard, which were all conducted in laboratory settings. These discrepancies have been shown for GARMIN_RP_ in cadence, vertical oscillation, and contact time [8,15,16] and for Stryd^TM^ in power and vertical oscillation [7,15]. However, other studies suggest good validity for GARMIN_RP_ in cadence, vertical oscillation, and contact time [4], along with good validity for Stryd^TM^ in contact time [7]. Nevertheless, it is crucial to highlight the lack of a gold standard for measuring running biomechanics variables in a real trail running environment, which is a general limitation of the research. Despite the lack of a universal gold standard, our research provides valuable insights, as we offer direct comparisons between devices, aiding users in understanding their consistency and accuracy. Additionally, our findings could guide device selection for athletes and coaches, contribute to ongoing research advancements, and highlight areas for device improvement. Overall, our study enhances the understanding of and decision-making in trail running biomechanics, despite the absence of a gold standard.

### 4.2. Reliability and Validity of GARMIN_RP_

Extensive research has been conducted on GARMIN_RP_, shedding light on its reliability and validity across several mechanical running metrics. Adams et al. [4] researched the monitoring capabilities of the GARMIN_RP_ Fenix 2 paired with a heart rate strap as a reliable and cost-efficient tool for assessing the dynamics of running, but tests were carried out in laboratory settings using a treadmill. The authors compared the GARMIN_RP_ watch against the gold standard motion-capture system and an instrumented treadmill to collect ground reaction force data across three conditions: baseline (self-selected speed and cadence), higher cadence, and decreased vertical motion (minimal oscillation). Positioned as a practical substitute for expensive laboratory configurations, the watch adeptly detects variations in cadence, vertical oscillation, and ground contact time when there are changes in running patterns. Noteworthy is the positive impact of cadence adjustments on biomechanics, leading to a reduction in elements linked to potential injuries [25]. On the other hand, ground contact time showed lower validity compared to cadence and vertical oscillation, but despite this, it still showcases commendable dependability, endorsing its application in the context of gait retraining. Similarly, another recent study showed that the ground contact time reported by GARMIN_RP_ in their study significantly differed from the gold standard (bilateral force plate) at all rates, underestimating it [16]. These discrepancies reported in ground contact times during running in place emphasize the context-specific considerations needed when assessing wearable device performance, as the biomechanical differences are too extensive to allow the measurement of contact time using the same methodology [16]. This aligns with the findings of the current study, where, despite a high reliability being observed in various variables, power and contact time emerged with the lowest reliability. Additionally, concerning inter-device reliability, the variables that demonstrated lower reliability in our analysis were uphill power, vertical oscillation, and contact time. As it is shown in Figure 3a,b, the higher the power (both uphill and downhill), the higher the distance to the mean is. That reveals a systematic error in this metric. There was also a clear bias when assessing vertical oscillation (Figure 3e,f) where the error detected was always positive (higher than 0), suggesting that both devices detect different values, with the GARMIN_RP_ values always higher. The last metric worth mentioning is ground contact time (Figure 3h). It can be seen that is had a large error when running downhill (but not uphill), which may affect the other outputs that were already mentioned if these data are incorporated into the mathematical model of each device’s calculations.

### 4.3. Reliability and Validity of Stryd^TM^

Three studies been conducted on Stryd^TM^ [6,7,9], studying its reliability and validity across various biomechanical running parameters. It is noteworthy that most of the research on Stryd^TM^ compares the measurements obtained by this device with those of other validated devices or with the gold standard. The intra-device reliability, as indicated by the coefficient of variation, for variables including contact time, flight time (closely linked to ground contact time), and cadence obtained from the Stryd^TM^ device indicates its adequacy in treadmill running assessments, allowing for the effective monitoring and tracking of functional performance changes over time [9].

Addressing the validity of Stryd^TM^, discrepancies were observed after comparing the measurements obtained from this wearable device with those from the gold standard. These disparities in assessing power with the Stryd^TM^ Power Meter and with the force platforms indicated a consistent trend, implying a proportional error in the Stryd^TM^ power estimation as the running speed increased [7]. The potential cause of this underestimation in power meter readings may be linked to the utilization of the apparent mechanical efficiency in the power calculation process. While mechanical efficiency assessment is a key focus of running power meters, the method of its integration into power output estimation appears to contribute to significant variations in absolute power values [7]. Notably, the Stryd^TM^ team referenced a gross mechanical efficiency of 25%, achievable by elite runners, which represents approximately half of the apparent mechanical efficiency values reported by other researchers and those identified in our study [7]. The substantial difference in mechanical efficiency values could account for the observed underestimation of power output by the Stryd^TM^ Power Meter when compared to established reference systems [7].

Expanding on the discussion of power metrics, the primary objective of the study performed by Cartón-Llorente et al. [19] was to evaluate the reliability and agreement between the Stryd^TM^ and RunScribe^TM^ systems in measuring running power on a treadmill. Despite both systems demonstrating dependable power output data and a near-perfect correlation, caution is warranted when using them interchangeably due to the observed inconsistencies. Both devices showed a high absolute reliability, with Stryd^TM^ exhibiting more reliability than RunScribe^TM^, consistent with the findings from a study by Cerezuela-Espejo et al. [11]. However, wide variations in agreement limits and a substantial random error were reported by Cartón-Llorente et al. [19], possibly attributed to methodological differences, including the disparate sampling rates (1000 Hz for Stryd vs. 500 Hz for RunScribe) and algorithmic variations. Taboga et al. [26] also compared the running powers and various biomechanical parameters measured by different devices to force-plate measurements and provided insights into the accuracy and reliability of GARMIN_RP_ and Stryd^TM^. The authors found that compared to force-based measurements obtained from a treadmill equipped with force plates, both GARMIN_RP_ and Stryd^TM^ showed differences in their measurements of running power. Specifically, Garmin was found to overestimate the step length and consequently running speed, while Stryd^TM^ underestimated both the step length and step frequency, leading to a slower reported speed. These differences in measurements resulted in distinct biases in running power compared to the force-based measurements [26]. The lack of algorithm disclosure by the companies complicates direct comparisons, emphasizing the need for standardized definitions in running power assessments.

The article by Aubry et al. [27] focused on the relationship between running power and metabolic demand, particularly utilizing Stryd^TM^ to measure running power. Despite an overall significant relationship being established, it was notably weak (r = 0.29), suggesting that running power, as assessed by the Stryd^TM^ Power Meter, may not accurately reflect the metabolic demand in a diverse population of runners. A further analysis revealed that individual mechanical components, such as vertical oscillation and ground contact time, provided more valuable predictive information for metabolic demand than running power, with significant relationships found in recreational runners but not in elite runners. The study also explored the impact of running surface on metabolic demand, indicating that overground running was more metabolically costly than treadmill running for almost all speeds, with the difference likely attributed to surface stiffness and elasticity. These findings emphasize the relationship between running power, metabolic demand, and surface type, cautioning against the simplistic use of running power as a surrogate for metabolic demand or for assessing running economy. This underscores the importance and necessity of studies conducted in real-world natural settings, such as the current article.

### 4.4. Stryd^TM^ vs. GARMIN_RP_

In the present study, the results from investigating the intra-device reliability followed a similar trend, revealing broader limits of agreement and mean bias for power compared to the other variables in the Bland–Altman analyses, with coefficients of variation indicating a higher measurement variability for power. In the analysis of inter-device reliability, this trend became more apparent, with wider limits of agreement and mean bias, suggesting a higher degree of systematic bias between the measurements obtained from GARMIN_RP_ vs. Stryd^TM^ in power, especially on positive slopes.

On the other hand, in a publication authored by García-Pinillos et al. [9], cadence showed consistent validity (less than a 1% difference) across the entire range of running velocities, indicating reliability when compared with the OptoGait system [9]. Likewise, according to Pinedo-Jauregi et al. [6], validity measures for cadence were acceptable during constant walking on positive slopes and level walking with load carriage.

Lastly, regarding the ground contact time, García-Pinillos et al. [9] reported an underestimation and poor reliability for the measurement of ground contact time when comparing a power meter to an OptoGait system. The paired comparisons emphasized that Stryd^TM^ tends to underestimate contact time (up to 8%) and overestimate flight time (up to 67%) compared to OptoGait, particularly at lower running velocities [9]. In contrast, another study revealed an underestimation of ground contact time by the Stryd^TM^ Power Meter, although the discrepancy was deemed negligible when compared to readings from force platforms [7]. Pinedo-Jauregi et al. [6] also investigated the performance of the Stryd^TM^ Power Meter in measuring ground contact time under various walking conditions on different positive slopes and with varying backpack loads. The Stryd^TM^ Power Meter demonstrated reliability in measuring ground contact time but a systematic bias in overestimating ground contact time under both constant walking on positive slopes and level walking with a loaded backpack [6]. The observed discrepancies in temporal variables, particularly ground contact time, may arise from inherent modeling aspects of Stryd’s vertical ground reaction forces, potentially leading to the omission of passive peaks [7]. Additionally, these differences are hypothesized to be partially linked to the height disparity of OptoGait’s LEDs, impacting the timing of events such as heel contact and toe lift-off within the gait cycle [9]. Moreover, the accuracy of the OptoGait system might not be as reliable as the force platform reference system chosen by Imbach et al. [7], providing a possible explanation for the contrasting outcomes. Pinedo-Jauregi et al. [6] suggested that factors beyond positioning may contribute to the inaccuracies in Stryd’s contact time measurements, and one plausible explanation could be linked to the distinct algorithms employed by the manufacturers for quantifying contact time in each exercise mode.

Only two studies to date have compared biomechanical measurements of running obtained from wearable devices such as GARMIN_RP_ and Stryd^TM^, with a focus on vertical oscillation [15] and cadence [8]. Smith et al. [15] examined the validity and reliability of vertical oscillation measurements across diverse devices, including GARMIN_RP_ and Stryd^TM^, comparing them to video analysis as the gold standard. The findings mirrored those of our study, revealing strong intra-device reliability, supported by robust correlation coefficients. Nevertheless, a notable divergence among devices was evident, with Stryd^TM^ displaying underestimation and GARMIN_RP_ exhibiting overestimation of vertical oscillation. While wearable devices offer credible and dependable measures in vertical oscillation, variations in values between devices emphasize the non-comparability of the data reported by GARMIN_RP_ and Stryd^TM^. It is essential to note that the treadmill-based tests used in this article warrant consideration regarding the potential influence of treadmill running on vertical oscillation when extrapolating findings to outdoor running. Similarly, the results reported in the present article showed that vertical oscillation during uphill ascent exhibited the highest coefficient of variation. Moreover, the results obtained when comparing the measurements of GARMIN_RP_ and Stryd^TM^ followed a similar trend, showing notably wide limits of agreement and mean bias for vertical oscillation in both the uphill and downhill Bland–Altman analyses, and the coefficients of variation were particularly high compared to other variables, suggesting higher measurement variability for vertical oscillation, especially in downhill analyses.

The findings from previous scientific studies align with the results of the current study, where the graphical representations revealed distinct trends indicative of potential errors. In Figure 4a, a substantial difference in power is evident, suggesting a significant overestimation in Garmin’s power measurements compared to Stryd. Figure 4b displays a noteworthy discrepancy in speed, indicating a substantial difference in speed measurements between the two devices. Vertical oscillation, portrayed in Figure 4c, showed a clear divergence, with Garmin showing underestimated values compared to Stryd. Finally, Figure 4d reveals a significant difference, suggesting a distinct variation in Garmin’s measurements compared to those of Stryd. These trends in error underscore the importance of scrutinizing and addressing device-specific variations to enhance the accuracy and reliability of the measured metrics in the context of athletic performance monitoring systems.

On the other hand, a recent study assessing the validity of cadence measurements across various devices reported notable findings [8]. Specifically, during running in place tests, the GARMIN_RP_ Running Pod stood out as the sole device exhibiting significant differences in step rate measurements—both at medium and high intensity—compared to the gold standard force plates. This suggests a need for caution when interpreting cadence data reported by Garmin, emphasizing the importance of considering potential variations in measurement outcomes under different testing conditions.

### 4.5. Practical Application

However, although biomechanical variables seem to differ depending on whether running is performed on a treadmill or in real trail running conditions, these variables appear to behave similarly based on speed, as Lang et al. [28] showed in the results of their article. Despite a higher cadence and shorter ground contact times on the treadmill compared to trail running, the downhill cadence increased with speed and ground contact times decrease similarly in both settings [28]. However, vertical oscillation diverged; it decreased with speed in laboratory tests, and it increased on negative slopes in trail running where the speed was higher [28]. This discrepancy is likely attributed to the terrain slope, which significantly impacts biomechanical and physiological adaptation, alongside adjustments in the center of gravity during uphill and downhill running.

It is worth noting the potential impact of terrain and environmental conditions on the accuracy of wearable devices, as demonstrated in a recent article by Uwe Schlink [29]. This article sheds light on the challenges and applications of wearable sensors in diverse environmental conditions and terrains, emphasizing the influence of recording intervals on sensor performance. The authors emphasized the necessity of investigating the accuracy of running metrics measurements obtained with wearable devices in different terrains and environmental conditions. This can be achieved by testing the devices in the specific context of trail running sports modalities to evaluate the intra-device reliability of the wearable devices. The results obtained in the present research revealed discrepancies between the measurements obtained with Stryd^TM^ and GARMIN_RP_. These observed variations may be attributed to inherent differences in the algorithms, sensor technologies, or calibration methods employed by both devices.

Lastly, wearable technology, as evidenced by some previous evidence [5,12,30,31], hold promise in predicting and preventing injuries, enhancing sports biomechanics, and addressing musculoskeletal concerns, emphasizing the need for robust methodologies and clear reporting in research. In addition, real-time monitoring of biomechanical variables in trail running has emerged as a promising future perspective. This innovation would provide runners with the ability to receive instant feedback on their technique, enabling personalized adjustments during the run. Furthermore, the early identification of inefficient biomechanical patterns could contribute to injury prevention, while optimizing performance on changing terrains would be possible through immediate adjustments. Advances in technologies such as wearable sensors and cloud-based data analysis would facilitate the implementation of this real-time monitoring, not only enhancing individual performance but also contributing to scientific research and the development of innovative strategies in the field of trail running [30,31]. Moreover, integrating mechanical, vertical, and metabolic data into apps and smartwatches enhances real-time insights for trail runners, allowing immediate adjustments and a more personalized, data-driven approach to optimize performance. Nevertheless, measuring biomechanical variables in trail running faces challenges due to the absence of a universal gold standard. Wearable devices like inertial sensors and GPS trackers offer insights, but the lack of standardized approaches raises concerns about data reliability. Further research is crucial to refine methodologies, address individual biomechanical variations, and enhance the validity of measurements in trail running using existing wearable technologies.

Our research approach stands out for its innovative stride into addressing the existing gap in the biomechanical evaluation of trail running under real-world conditions. While past studies have primarily relied on laboratory settings, our research took a pioneering step by harnessing the power of recent technological advancements in wearable devices. By exploring the reliability and consistency of GARMIN_RP_ and Stryd^TM^ devices in measuring key biomechanical variables during trail running, including analyses for both uphill and downhill scenarios, our research provides a novel contribution to the field, offering valuable insights with practical implications for both researchers and practitioners in trail running and sports science.

### 4.6. Limitations and Strengths

Several limitations and strengths need to be mentioned in this article. The main limitation is the lack of biomechanical running measurements through gold standard devices to assess the agreement of these measurements with the wearable devices from GARMIN_RP_ and Stryd^TM^. In addition, this article included only two wearable devices, excluding other commercially available devices such as RunScribe that could have enriched this comparative analysis. Moreover, the small sample size and variations among participants, including differences in gender and fitness levels, represent another noteworthy limitation. This introduces the possibility of outliers and suggests that some device discrepancies may be partly attributed to participant variations. The primary strength of this study lies in conducting tests under real-world conditions: running outdoors. Another strength is the comparison of two devices of the same model, both from GARMIN_RP_ and Stryd^TM^, to explore the intra-device validity.

## 5. Conclusions

In conclusion, our study presented comprehensive findings on the intra- and inter-device reliability of GARMIN_RP_ and Stryd^TM^ devices in assessing various running metrics during trail running. The intra-device reliability analysis revealed a high consistency and agreement for both devices, with strong correlations, narrow limits of agreement, and low measurement variability across all variables. However, distinctions emerged in the inter-device reliability, particularly in uphill power and contact time and downhill vertical oscillation, indicating potential variations between GARMIN_RP_ and Stryd^TM^ measurements for certain running metrics. These noted discrepancies could be linked to inherent distinctions in the algorithms, sensor technologies, or calibration approaches utilized by each device.

These findings align with those of the existing literature, highlighting the need for continued research in natural field conditions. Overall, our research contributes valuable insights into the reliability of wearable devices during trail running, emphasizing the importance of considering contextual factors, such as the type of device or terrain level (uphill and downhill sections), in assessing the performance of different athletes or on different tracks.

## Figures and Tables

**Figure 1 sensors-24-03570-f001:**
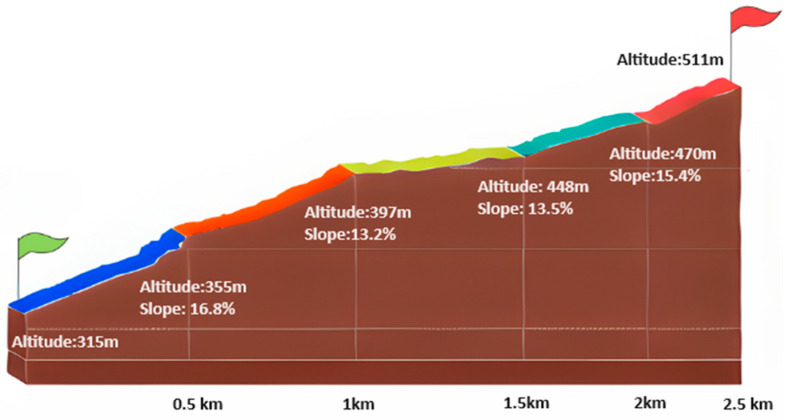
Trail course profile.

**Figure 2 sensors-24-03570-f002:**
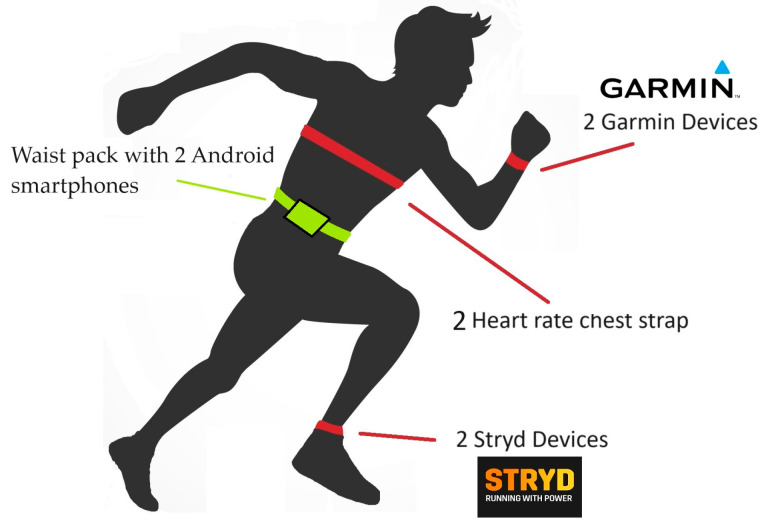
Graphical description of runner instrumentation with wearable electronic devices for the test.

**Figure 3 sensors-24-03570-f003:**
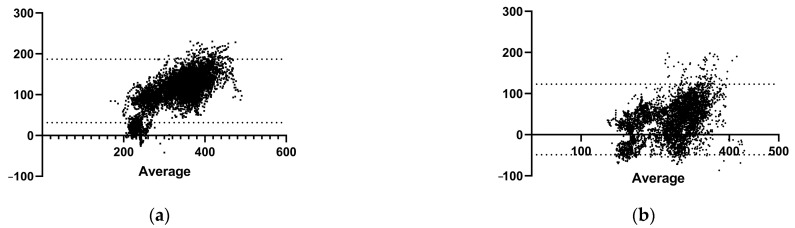
Bland–Altman plots of the rate of running in place as measured with (**a**) uphill power (W), (**b**) downhill power (W), (**c**) uphill speed (km/h), (**d**) downhill speed (km/h), (**e**) uphill vertical oscillation (cm), (**f**) downhill vertical oscillation (cm), (**g**) uphill contact time (ms), (**h**) downhill contact time (ms). The dashed lines represent the limits of agreement (LoA), while the solid lines depict bias.

**Figure 4 sensors-24-03570-f004:**
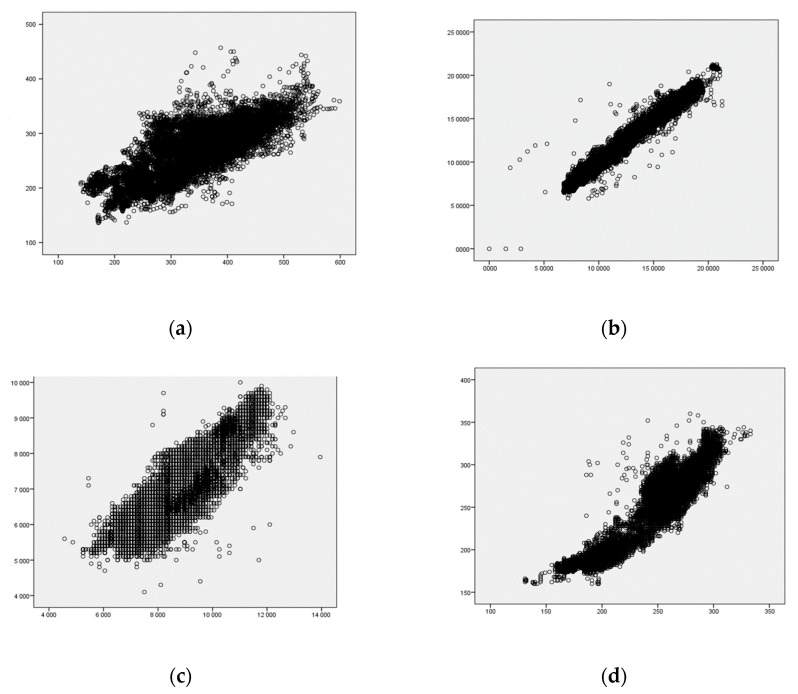
Univariate linear regression analysis of (**a**) power (W); (**b**) speed (km/h); (**c**) vertical oscillation (cm); (**d**) contact time (ms). The *y*-axis represents GARMIN_RP_ measurements, while the *x*-axis represents Stryd^TM^ measurements. The units are consistent with the measurements.

**Table 1 sensors-24-03570-t001:** Descriptive data for GARMIN and Stryd devices.

	Power (W)	Speed (km/h)	Cadence (Steps per Minute)	Vertical Oscillation (cm)	Contact Time (ms)
Garmin	339.4 ± 84.0	11.6 ± 3.3	86.9 ± 2.6	8.7 ± 1.5	248.6 ± 35.1
Uphill	385.1 ± 71.4	9.5 ± 1.3	85.6 ± 1.9	8.2 ± 1.2	267.9 ± 19.7
Downhill	299.8 ± 65.4	15.5 ± 2.3	88.6 ± 2.4	9.9 ± 1.2	210.5 ± 27.1
Stryd	267.3 ± 48.8	11.0 ± 4.1	86.3 ± 3.7	7.0 ± 1.1	255.0 ± 68.9
Uphill	272.6 ± 44.8	9.3 ± 1.3	85.5 ± 1.8	6.4 ± 0.7	279.8 ± 28.6
Downhill	256.5 ± 49.3	15.3 ± 2.2	88.4 ± 2.4	7.9 ± 0.9	206.1 ± 24.1

**Table 2 sensors-24-03570-t002:** Comprehensive summary of reliability outcomes for GARMIN_RP_ and Stryd^TM^ devices in running biomechanics evaluation.

		ICC (95%CI)	CV (%)	Bland–Altman Analysis		Pearson Correlation	Univariate Linear Regression Analysis	
				Mean Bias	LoA		B	t
GARMIN_RP_								
Power (W)	WT	0.990 (0.989–0.991)	2.8 ± 2.9	−2.9 ± 16.9	−36.1–30.2	0.981 *	B = [1.0]	t = [457.9] *
	U	0.993 (0.992–0.993)	1.9 ± 1.6	0.2 ± 13.1	−25.4–25.9	0.986 *	B = [0.993]	t = [417.8] *
	D	0.927 (0.911–0.940)	3.3 ± 2.8	−5.5 ± 20.5	−45.7–34.6	0.872 *	B = [0.871]	t = [71.5] *
Speed (km/h)	WT	0.997 (0.997–0.997)	1.4 ± 3.1	−0.004 ± 0.4	−0.7–0.7	0.993 *	B = [0.987]	t = [863.7] *
	U	0.987 (0.986–0.988)	1.3 ± 1.7	0.009 ± 0.3	−0.7–0.7	0.975 *	B = [0.985]	t = [337.6] *
	D	0.993 (0.992–0.994)	1.1 ± 1.5	−0.05 ± 0.4	−0.8–0.7	0.987 *	B = [1.0]	t = [282.9] *
Cadence	WT	0.978 (0.977–0.978)	0.3 ± 0.5	0.001 ± 0.8	−1.5–1.5	0.956 *	B = [0.969]	t = [323.3] *
	U	0.964 (0.962–0.966)	0.3 ± 0.5	−0.02 ± 0.7	−1.4–1.4	0.931 *	B = [0.939]	t = [197.3] *
	D	0.979 (0.978–0.981)	0.3 ± 0.4	0.02 ± 0.7	−1.3–1.4	0.960 *	B = [0.974]	t = [158.0] *
Vertical oscillation	WT	0.982 (0.978–0.985)	2.3 ± 2.5	−0.1 ± 0.4	−0.8–0.6	0.967 *	B = [0.998]	t = [348.9] *
	U	0.973 (0.966–0.978)	2.4 ± 2.6	−0.1 ± 0.4	−0.8–0.6	0.951 *	B = [0.978]	t = [220.2] *
	D	0.973 (0.970–0.976)	1.6 ± 1.6	0.003 ± 0.3	−0.7–0.7	0.954 *	B = [1.1]	t = [127.1] *
Contact time	WT	0.990 (0.987–0.993)	1.5 ± 1.5	−2.2 ± 6.4	−14.9–10.4	0.984 *	B = [1.027]	t = [504.2] *
	U	0.971 (0.961–0.977)	1.3 ± 1.1	−2.0 ± 6.0	−13.7–9.8	0.953 *	B = [1.1]	t = [225.1] *
	D	0.967 (0.942–0.979)	1.8 ± 1.8	−2.8 ± 5.9	−14.4–8.8	0.959 *	B = [1.1]	t = [135.4] *
	STRYD^TM^							
Power (W)	WT	0.980 (0.978–0.983)	2.4 ± 2.5	2.6 ± 12.7	−22.3–27.4	0.963 *	B = [0.935]	t = [227.4] *
	U	0.981 (0.980–0.982)	2.1 ± 2.06	0.3 ± 11.6	−22.4–23.0	0.963 *	B = [0.942]	t = [216.2] *
	D	0.983 (0.965–0.990)	2.6 ± 2.2	5.8 ± 10.9	−15.6–27.1	0.974 *	B = [0.948]	t = [197.2] *
Speed (km/h)	WT	0.997 (0.997–0.997)	1.5 ± 4.0	−0.04 ± 0.4	−0.8–0.7	0.993 *	B = [0.987]	t = [727.7] *
	U	0.991 (0.990–0.992)	1.2 ± 1.2	−0.04 ± 0.2	−0.5–0.4	0.984 *	B = [0.949]	t = [333.1] *
	D	0.990 (0.989–0.991)	1.6 ± 1.3	−0.05 ± 0.4	−0.9–0.8	0.980 *	B = [0.965]	t = [226.9] *
Cadence	WT	0.982 (0.980–0.984)	0.4 ± 0.5	−0.1 ± 0.8	−1.6–1.3	0.966 *	B = [0.968]	t = [289.9] *
	U	0.954 (0.948–0.959)	0.4 ± 0.5	−0.1 ± 0.7	−1.5–1.2	0.915 *	B = [0.886]	t = [136.7] *
	D	0.977 (0.974–0.980)	0.4 ± 0.5	−0.1 ± 0.7	−1.5–1.3	0.957 *	B = [0.950]	t = [152.3] *
Vertical oscillation	WT	0.973 (0.961–0.980)	2.3 ± 2.6	−0.1 ± 0.3	−0.8–0.5	0.954 *	B = [0.945]	t = [244.5] *
	U	0.903 (0.834–0.937)	2.8 ± 2.9	−0.2 ± 0.3	−0.9–0.5	0.856 *	B = [0.939]	t = [158.5] *
	D	0.987 (0.984-.989)	1.478 ± 1.1	−0.05 ± 0.2	−0.5–0.4	0.978 *	B = [0.905]	t = [216.6] *
Contact time	WT	0.993 (0.988–0.995)	1.9 ± 2.1	4.1 ± 9.8	−15.1–23.3	0.988 *	B = [0.998]	t = [484.9] *
	U	0.957 (0.914–0.974)	2.0 ± 2.2	5.5 ± 10.2	−14.6–25.6	0.935 *	B = [0.777]	t = [99.5] *
	D	0.985 (0.976–0.990)	1.7 ± 1.3	2.3 ± 5.5	−8.5–13.0	0.975 *	B = [0.905]	t = [216.6] *

ICC = Intraclass Correlation Coefficient, CV = Coefficient of Variation, LoA = Limits of Agreement, WT = whole test, U = uphill, D = downhill. 95%CI = confidence interval with lower and upper limits. Significance was set at *p* < 0.001 (*).

**Table 3 sensors-24-03570-t003:** Comprehensive summary of reliability outcomes comparing GARMIN_RP_ vs. Stryd^TM^ devices in running biomechanics evaluation.

		ICC (95%CI)	CV (%)	Bland–Altman Analysis		Pearson Correlation	Univariate Linear Regression Analysis	
				**Mean Bias**	**LoA**		**B**	**t**
Power (W)	WT	0.534 (−0.216–0.802)	18.8 ± 9.0	82.2 ± 54.6	−24.9–189.3	0.764 *	B = [1.4]	t = [141.0] *
	U	0.444 (−0.116–0.779)	3.0 ± 3.2	109.1 ± 39.7	31.4–186.9	0.874 *	B = [1.4]	t = [168.4] *
	D	0.734 (0.272–0.869)	6.5 ± 110.8	37.0 ± 43.9	−49.0–123	0.743 *	B = [1.0]	t = [80.8]
Speed (km/h)	WT	0.991 (0.989–0.992)	0.5 ± 0.5	0.1 ± 0.6	−1.1–1.4	0.983 *	B = [0.991]	t = [664.2] *
	U	0.953 (0.940–0.962)	16.2 ± 6.2	0.2 ± 0.5	−0.9–1.2	0.919 *	B = [0.964]	t = [229.7] *
	D	0.972 (0.969–0.974)	3.9 ± 2.9	0.1 ±0.7	−1.3–1.6	0.948 *	B = [1.0]	t = [227.2]
Cadence	WT	0.969 (0.966–0.971)	0.5 ± 0.6	0.1 ± 0.9	−1.6–1.8	0.941 *	B = [0.949]	t = [347.5] *
	U	0.941 (0.938–0.944)	17.1 ± 7.0	0.09 ± 0.9	−1.6–1.8	0.891 *	B = [0.916]	t = [193.3] *
	D	0.958 (0.952–0.962)	3.9 ± 3.2	0.2 ± 0.1	−1.7–2	0.922 *	B = [0.922]	t = [181.6]
Vertical oscillation	WT	0.580 (−0.147–0.855)	11.9 ± 7.1	1.8 ± 0.8	0.3–3.3	0.857 *	B = [1.2]	t = [198.4] *
	U	0.370 (−0.165–0.704)	2.6 ± 2.6	1.8 ± 0.8	0.2–3.4	0.728 *	B = [1.3]	t = [99.6] *
	D	0.483 (−0.108–0.807)	−13.3 ± 14.4	1.9 ± 0.7	0.6–3.2	0.860 *	B = [1.2]	t = [123.0] *
Contact time	WT	0.950 (0.945–0.954)	0.6 ± 0.5	−3.2 ± 17.1	−36.7–30.3	0.923 *	B = [0.766]	t = [286.6] *
	U	0.790 (0.604–0.871)	14.7 ± 4.6	−9.8 ± 16.9	−43.0–23.3	0.735 *	B = [0.582]	t = [101.7] *
	D	0.917 (0.794–0.956)	4.1 ± 2.5	7.6 ± 11.8	−15.6–30.7	0.905 *	B = [1.1]	t = [154.6] *

ICC = Intraclass Correlation Coefficient, CV = Coefficient of Variation, LoA = Limits of Agreement, WT = whole test, U = uphill, D = downhill. 95%CI = confidence interval with lower and upper limits. Significance was set at *p* < 0.001 (*).

## Data Availability

The data presented in this study are available on request from the corresponding author.

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
