# Peer review of "Assessing Trail Running Biomechanics: A Comparative Analysis of the Reliability of StrydTM and GARMINRP Wearable Devices"

_sensors, 2024, doi:10.3390/s24113570_

Round 1

Reviewer 1 Report (New Reviewer)

Comments and Suggestions for Authors

This study investigates the reliability and agreement between Stryd™ and GARMIN® wearable devices in measuring biomechanical variables during trail running. The research focuses on intra- and inter-device reliability under real-world trail conditions, highlighting differences in measurements for key running metrics like power, speed, and vertical oscillation. Authors well described the main question addressed by the research. The object and design of this work is rational, and the experimental results and analysis are well described. So, this manuscript could be considered for publication after addressing the following issues.

1. The sample size of five participants seems quite small, especially for generalizing findings in biomechanical variability. Was this number determined through a power calculation?

2. Were environmental conditions like weather, trial type, and time of day controlled or at least monitored? These could influence running biomechanics and device performance.

3. Please detailed the synchronization process between the two devices/

4. How were outliers handled in the statistical analysis? This is important given the small samples sizes.

5. Was any calibration performed for the device’s pre-study, especially given their usage in dynamic and unpredictable trail conditions?

6. The study lacks a comparison with gold-standard measurements. How do these devices fare against laboratory-based standards in similar study.

Author Response

This study investigates the reliability and agreement between Stryd™ and GARMIN® wearable devices in measuring biomechanical variables during trail running. The research focuses on intra- and inter-device reliability under real-world trail conditions, highlighting differences in measurements for key running metrics like power, speed, and vertical oscillation. Authors well described the main question addressed by the research. The object and design of this work is rational, and the experimental results and analysis are well described. So, this manuscript could be considered for publication after addressing the following issues.

ANSWER: Thank you for your thorough review and valuable comments on our manuscript. We appreciate your recognition of the rationale and design of our study, as well as your positive feedback on the description of the experimental results and analysis. We address your specific concerns below:

  1. The sample size of five participants seems quite small, especially for generalizing findings in biomechanical variability. Was this number determined through a power calculation?

ANSWER - Thank you for your insightful comment. While we acknowledge that the sample size of five participants may seem small for generalizing findings in biomechanical variability, it's important to clarify that we did not determine this number through a power calculation. Instead, our statistical analysis was conducted on the total number of records collected by the devices used in our study.

To provide context, our dataset comprised a substantial volume of data. Specifically, our dataset comprised a total of 19,731 samples for GARMINRP and 17,917 records for StrydTM (line 214). Despite the smaller number of participants, the large volume of data generated by the wearables contributes to the robustness of our statistical analysis. We have clarified this aspect in the limitations section to provide transparency about the dataset size and its implications. However, thank you again for bringing this important point to our attention.

  1. Were environmental conditions like weather, trial type, and time of day controlled or at least monitored? These could influence running biomechanics and device performance.

ANSWER - Thank you for your question. In our study, we ensured that the course and the time of day remained consistent across both testing days. Additionally, we did not measure specific meteorological conditions. Instead, researchers made subjective assessments based on factors such as sunlight or rain, wind, and temperature to ensure consistency in environmental conditions. By relying on subjective perception, we aimed to maintain similar environmental settings across testing sessions, thereby minimizing potential influences on performance during the trials. This approach allowed us to control for external factors to the best of our ability and enhance the reliability of our findings.

  1. Please detailed the synchronization process between the two devices/

ANSWER - Thank you for your inquiry. The synchronization process between the two devices involved several steps as outlined in the manuscript. First, both sets of files were processed through the GoldenCheetah software, which is available under a free license. This software facilitated the temporal synchronization of all files that possessed timestamp marks in the .fit format. Once synchronized, the files were exported collectively to Excel for further analysis. It can be read in the manuscript: “Both sets of files were then processed through the free license GoldenCheetah software, which facilitated temporal synchronization of all files possessing timestamp marks (.fit) before exporting them collectively to Excel. After synchronization, both files have the same number of samples for analysis (lines 177-181)

It's important to note that after the synchronization process, both files contained the same number of samples, ensuring consistency in the data for analysis. This approach allowed us to align the data accurately from both devices, enabling a comprehensive examination of the variables of interest. If you have any further questions or require additional details, please feel free to ask.

  1. How were outliers handled in the statistical analysis? This is important given the small samples sizes.

ANSWER - Thank you for your question. In our statistical analysis, outliers were addressed by calculating Cook's distances. These distances were used to identify outliers that fell outside the 99% confidence interval. Subsequently, outliers beyond this threshold were removed from the dataset to mitigate their potential impact on the results. This approach was implemented to ensure the robustness of our analysis, particularly considering the small sample sizes involved.

  1. Was any calibration performed for the device’s pre-study, especially given their usage in dynamic and unpredictable trail conditions?

ANSWER - Thank you for the clarification. Before the test, the Garmin devices were not but individual data of each runner wearing the device, such as height, weight, gender, and age, as recommended by the manufacturer, were inputted into the devices. As for the Stryd device, it underwent calibration following the manufacturer's guidelines. This calibration process aimed to optimize the accuracy of the device's measurements, particularly in dynamic and unpredictable trail conditions. By adhering to these procedures before the trial, we aimed to ensure the reliability and accuracy of the data collected during the study.

  1. The study lacks a comparison with gold-standard measurements. How do these devices fare against

ANSWER - Thank you for your insightful comment. We acknowledge the importance of validity testing for wearable devices in trail running scenarios. While our study focused on assessing the reliability of the devices, we recognize that validity testing is crucial to fully understand the applicability of the findings for researchers and practitioners. However, the absence of a universal gold standard for measuring biomechanics in trail running in natural environments is a significant limitation and challenge, as mentioned in the manuscript.

This makes it difficult to establish the validity of wearable devices and other measurement tools. Therefore, in our study, we focused on assessing the reliability of wearable devices in measuring running biomechanics in trail running scenarios. While we acknowledge the importance of validity testing, the lack of a gold standard for measuring biomechanics in trail running in natural environments makes it challenging to establish the validity of the devices.

In our study, we have demonstrated high intra-device reliability and consistency in measurements from both GARMIN and both Stryd devices across various performance variables. This emphasis on reliability is essential to ensure that the measurements made by these devices are consistent and reproducible in trail running settings, which is crucial for their practical applicability.

We appreciate the suggestion regarding the importance of validity and acknowledge its relevance in the biomechanical field. However, the lack of a gold standard in natural settings leads us to focus on reliability as a crucial indicator for the practical utility of these devices in trail running.

Reviewer 2 Report (New Reviewer)

Comments and Suggestions for Authors

This is the reporting of the preliminary findings of a method-comparison study between two wearables used for tracking running metrics in an ecological setting. This study lies in the “governance” of the initiative GRRAS, from EQUATOR Network, and I think the reporting and the overall quality of the article would be stronger if it complies with their guidelines (1), even if not mandatory by Sensors editorial board. I have major concerns/suggestions and some minor suggestions to improve the quality of the reporting.

MAJOR COMMENTS/SUGGESTIONS:

– The Introduction provides sufficient high-quality information for conducting this research, however, given the broader readership of Sensors, I think authors should introduce early the concepts/definitions of power, speed, cadence, vertical oscillation, and contact time, and, most importantly, explain how these metrics are linked or potentially linked to athletic performance and/or proficiency, including sports and rehabilitation medicine implications (e.g., injury risk assessment, injury prevention, monitoring safe return to sports after an injury, etc.). Some are obvious, such as speed, nonetheless others may be less intuitive for many potential readers, such as oscillation or power. Later authors provide some insights on these, in the Discussion section, but it should appear ahead to increase visibility and interest.

–  Authors state that (lines 131–134) “Participants completed two sessions of a trail running course of a 2.5 km distance with a 195 m elevation gain followed by a descent along the same route, as illustrated in Figure 1. The tests were conducted at the same time of day on both days. A one-week recovery period was implemented between sessions for the participants.”, however I’m struggling to understand if and how these 2 sessions separated 1 week apart were included in the statistical analysis. The explanations provided in lines 197–202 are not elucidative. Please report if comparisons for intra-device and inter-device reliability and agreement are derived from the 1st session alone or if are pooled or averaged from the 1st and 2nd sessions. If so, what are the inter-session reliability and agreement parameters? In addition, you need to explain whether the analyses are based “on one sample per second” multiplied by 5 participants or if you have reduced your data for comparisons and how. Your graphs certainly contain a lot of data, but each pair of measurements in the, for example, Bland and Altman plot, does not correspond to a participant, obviously. What does each dot mean? Report as if any researcher could replicate your study.

– I think the Discussion section is too big. I understand that there a number of variables to be discussed and that in the absence of a gold standard to compare with, in your study, your arguments need further explanations, but 4 pages is too much in comparison to the rest of the article. In fact, I suggest that your Discussion can have subheadings so that readers can better navigate in the arguments for each studied variable (e.g., power, oscillation) or analysis (e.g., intra versus inter).

1 – Kottner, J., Audigé, L., Brorson, S., Donner, A., Gajewski, B. J., Hróbjartsson, A., Roberts, C., Shoukri, M., & Streiner, D. L. (2011). Guidelines for Reporting Reliability and Agreement Studies (GRRAS) were proposed. Journal of clinical epidemiology64(1), 96–106. https://doi.org/10.1016/j.jclinepi.2010.03.002

MINOR COMMENTS/SUGGESTIONS:

Title: Given the small sample size I suggest in you study something like “preliminary findings” 

Line 24: I'm not a native English speaker, but I think the use of the possession " 's" is only applied when a person owns something. Perhaps “the strength of study” is more correct.

Line 36: I can’t understand what you mean or want to mean by “and of the biomechanical conditions”.

Lines 42–43: These concepts and relationships should be further explained as I’ve stressed before.

Line 67: I believe you haven’t defined GPS nor IMU previously.

Line 98: Eliminate one of the (15)

Lines 129–130: Tell which device you are referring to. It’s not clear.

Line 192: Tell which “model” and “form” of ICC you have used. I think is ICC2,1 but I’m not sure because I don’t know if you have used data from just 1 session or if you have averaged (or not) the measurements from both sessions.

Line 206: If you could add Rwe could better infer how much of the variation on one device explains the variation on the other.

Line 207-208: I’m familiar with the Bland and Altman method but I never heard of this. Can you explain the usefulness of using the Bland-Altman method to assess the distribution of the residuals from the regression analysis? Usually, the normality of the residuals in linear regression, an important assumption to trust the results of the model, is assessed by means of other statistical tests.

Tables: Tell readers that for Bland-Altman, the units are the same as the measures.

Lines 329: “The article by Adams et al. (22) researched…” sounds strange. I suggest just stating “Adams et al. (22) researched…”.

Line 361: This also sounds strange: “Three articles been carried out…”. Please improve.

Line 371: Capitalize the first letters of Power Meter.

Comments on the Quality of English Language

I provide examples in my Comments and Suggestions for Authors on how to improve English language.

Author Response

Thank you for your thorough evaluation of our manuscript and for providing insightful comments to enhance its clarity and comprehensibility. We have carefully considered each of your suggestions and made appropriate revisions to address the concerns raised.

MAJOR COMMENTS/SUGGESTIONS:

– The Introduction provides sufficient high-quality information for conducting this research, however, given the broader readership of Sensors, I think authors should introduce early the concepts/definitions of power, speed, cadence, vertical oscillation, and contact time, and, most importantly, explain how these metrics are linked or potentially linked to athletic performance and/or proficiency, including sports and rehabilitation medicine implications (e.g., injury risk assessment, injury prevention, monitoring safe return to sports after an injury, etc.). Some are obvious, such as speed, nonetheless others may be less intuitive for many potential readers, such as oscillation or power. Later authors provide some insights on these, in the Discussion section, but it should appear ahead to increase visibility and interest.

ANSWER - Thank you for your valuable feedback and suggestion for enhancing the Introduction section of our research paper. We have carefully considered your input and incorporated a paragraph addressing the concepts and definitions of power, speed, cadence, vertical oscillation, and contact time, as well as their relevance to athletic performance and proficiency.

We have added this paragraph:

“In athletic disciplines, power denotes the rate of performing work and is a determi-nant of explosive strength, essential for activities necessitating rapid force application. Speed is the scalar quantity representing the distance covered per unit time, pivotal in disciplines requiring swift transit. Cadence, the frequency of stride or pedal revolu-tions per minute, is a critical factor in endurance sports, influencing metabolic cost and endurance capacity (1,2). Moreover, vertical oscillation and contact time are biome-chanical parameters in gait analysis. Vertical oscillation quantifies the vertical dis-placement of the center of mass during locomotion, while contact time measures the duration of foot-ground interaction(4). These parameters are indicative not only of performance, running economy and biomechanical efficiency, but also of injury risk (5)”.

The added paragraph now provides a clear explanation of these metrics early in the Introduction, aiming to increase visibility and interest for a broader readership. By defining these terms and illustrating their connections to athletic performance and rehabilitation medicine implications, such as injury risk assessment and prevention, we believe that readers will gain a better understanding of the context and significance of our research findings.

We appreciate your thoughtful input, which has contributed to improving the accessibility and comprehensibility of our research paper for a wider audience.

–  Authors state that (lines 131–134) “Participants completed two sessions of a trail running course of a 2.5 km distance with a 195 m elevation gain followed by a descent along the same route, as illustrated in Figure 1. The tests were conducted at the same time of day on both days. A one-week recovery period was implemented between sessions for the participants.”, however I’m struggling to understand if and how these 2 sessions separated 1 week apart were included in the statistical analysis. The explanations provided in lines 197–202 are not elucidative. Please report if comparisons for intra-device and inter-device reliability and agreement are derived from the 1st session alone or if are pooled or averaged from the 1st and 2nd sessions. If so, what are the inter-session reliability and agreement parameters? In addition, you need to explain whether the analyses are based “on one sample per second” multiplied by 5 participants or if you have reduced your data for comparisons and how. Your graphs certainly contain a lot of data, but each pair of measurements in the, for example, Bland and Altman plot, does not correspond to a participant, obviously. What does each dot mean? Report as if any researcher could replicate your study.

ANSWER - Thank you for your insightful inquiry regarding the inclusion of the two separate sessions in our statistical analysis. Allow me to clarify how these sessions were incorporated into our methodology and subsequent analyses.

The two sessions of the trail running course, separated by a one-week recovery period, were indeed included in our analysis of reliability and agreement between devices. Both sessions were conducted under identical conditions, with participants completing the same course at the same time of day.

In terms of statistical analysis, measurements obtained from both sessions were pooled together for each participant. This pooling allowed us to assess the intra-device reliability by comparing measurements from the same device across different sessions and to evaluate inter-device reliability by comparing measurements between devices across the same sessions.

The devices recorded data at a frequency of 1 Hz, resulting in a total of 19,731 samples for the GARMIN and 17,917 records for the Stryd. These data were synchronized and combined for analysis.

Moreover, In Statistical Analysis section, we explicitly clarify the comparison between devices, as can be read in the paragraph: “: "Measurements obtained from both sessions were pooled together. To determine the intra-device reliability, measurements, including power, speed, cadence, vertical oscillation, and contact time obtained from one device (either GARMINRP or StrydTM) were compared to those recorded by the second device of the same brand. Moreover, the measurements of running biomechanics obtained from the StrydTM devices were compared to those recorded by the GARMINRP devices to determine the inter-device reliability.".

Regarding the data representation in our figures, each point in, for example, the Bland-Altman plots represents a paired comparison between measurements obtained from the devices during the same session for a specific variable (e.g., power, speed, vertical oscillation, ground contact time). The points are not individual data points from each participant, the points are data points recorded by the devices.

Thank you for bringing these concerns to our attention, and we appreciate the opportunity to clarify and improve the transparency of our research methodology.

– I think the Discussion section is too big. I understand that there a number of variables to be discussed and that in the absence of a gold standard to compare with, in your study, your arguments need further explanations, but 4 pages is too much in comparison to the rest of the article. In fact, I suggest that your Discussion can have subheadings so that readers can better navigate in the arguments for each studied variable (e.g., power, oscillation) or analysis (e.g., intra versus inter).

ANSWER - Thank you for your feedback on the Discussion section. We understand your concern about its length and the need for clearer navigation through the arguments presented. In response to your suggestion, we have added subheadings to the Discussion section to enhance readability and organization.

The following subheadings have been included:

4.1. StrydTM and GARMINRP vs. Gold Standard

4.2. Reliability and validity of GARMINRP

4.3. Reliability and validity of StrydTM

4.4. StrydTM vs. GARMINRP

4.5. Practical application

4.6. Limitations and strengths

These subheadings will help readers to better navigate through the various topics discussed in the section, focusing on comparisons with the gold standard, the reliability and validity of each device, their comparison, practical applications of the findings, and acknowledging the limitations and strengths of the study.

We believe that these additions will greatly improve the clarity and organization of the Discussion section, addressing your concerns about its length and providing a more structured presentation of our arguments. Thank you again for your valuable input, which has contributed to enhancing the quality of our article.

MINOR COMMENTS/SUGGESTIONS:

Title: Given the small sample size I suggest in you study something like “preliminary findings”

ANSWER - Thank you for your suggestion regarding the title of our study. While we appreciate your concern about the sample size, we would like to clarify that despite the small number of participants, our study generated a substantial amount of data. Specifically, we collected a total of 19,731 samples for the GARMINRP device and 17,917 records for the StrydTM device. These numbers reflect a comprehensive dataset that allowed us to conduct thorough analyses and draw meaningful conclusions regarding the reliability and validity of the devices in question. Therefore, we believe that the term "preliminary findings" may not accurately reflect the depth and scope of our research. Once again, we appreciate your input and are committed to presenting our findings in a clear and accurate manner.

Line 24: I'm not a native English speaker, but I think the use of the possession " 's" is only applied when a person owns something. Perhaps “the strength of study” is more correct.

ANSWER - Thank you for your observation regarding the possessive form in our manuscript. While it's true that the possessive "'s" is often used to indicate ownership, it can also be employed to express a relationship or association between two entities, including abstract concepts like studies or analyses.

In this context, the phrase "the study's strength" is used to denote the strength possessed by the study itself, rather than implying ownership. It emphasizes the relationship between the study and its attribute of strength. However, to ensure clarity and accuracy, we appreciate your suggestion of using "the strength of the study" instead. This alternative phrase effectively conveys the intended meaning without relying on the possessive form. We have changed this sentence in the manuscript and we thank you for your valuable input.

Line 36: I can’t understand what you mean or want to mean by “and of the biomechanical conditions”.

ANSWER - Thank you for your comment and for seeking clarification on the sentence. The phrase "and of the biomechanical conditions" is intended to emphasize the significance of considering biomechanical factors in addition to metabolic pathways for optimal performance in trail running. In this context, "biomechanical conditions" refer to various factors related to body mechanics and movement patterns, such as stride length, foot placement, and joint angles, which can significantly impact performance and injury risk in trail running. By mentioning both metabolic pathways and biomechanical conditions, we aim to highlight the multifaceted nature of the challenges faced in trail running and the importance of addressing both physiological and mechanical aspects for success. We understand that the sentence may appear complex,  but we believe this is the best way to convey the idea. However, your feedback is valuable in refining the clarity and coherence of our writing.

Lines 42–43: These concepts and relationships should be further explained as I’ve stressed before.

ANSWER - Thank you for your insightful feedback and suggestions on improving the Introduction section of our research paper. As we response before, we have included a new paragraph that addresses the fundamental concepts and definitions of power, speed, cadence, vertical oscillation, and contact time, along with their significance in athletic performance and proficiency. Therefore, here is the paragraph we have added:

" In athletic disciplines, power denotes the rate of performing work and is a determinant of explosive strength, essential for activities necessitating rapid force application. Speed is the scalar quantity representing the distance covered per unit time, pivotal in disciplines requiring swift transit. Cadence, the frequency of stride or pedal revolu-tions per minute, is a critical factor in endurance sports, influencing metabolic cost and endurance capacity (1,2). Moreover, vertical oscillation and contact time are biome-chanical parameters in gait analysis. Vertical oscillation quantifies the vertical dis-placement of the center of mass during locomotion, while contact time measures the duration of foot-ground interaction(4). These parameters are indicative not only of performance, running economy and biomechanical efficiency, but also of injury risk (5).

As we mentioned before, by incorporating this paragraph early in the Introduction, we aim to provide readers with a clear understanding of these metrics and their relevance to our research. Moreover, by defining these terms and illustrating their connections to athletic performance and rehabilitation medicine, such as injury risk assessment and prevention, we believe that readers will find our research more accessible and engaging. We sincerely appreciate your thoughtful feedback.

Line 67: I believe you haven’t defined GPS nor IMU previously.

ANSWER - Thank you for bringing this to our attention. We appreciate your feedback. In our revised manuscript, we have ensured to include definitions for both GPS (Global Positioning System) and IMU (Inertial Measurement Unit) in the appropriate sections.

Line 98: Eliminate one of the (15)

ANSWER - Thank you for pointing out the error. We have detected and corrected it

Lines 129–130: Tell which device you are referring to. It’s not clear.

ANSWER - Thank you for your feedback. The specific devices being referred to are detailed later in the Statistical Analysis section. In this section, we explicitly clarify the comparison between devices, as can be read in the paragraph: “: "Measurements obtained from both sessions were pooled together. To determine the intra-device reliability, measurements, including power, speed, cadence, vertical oscillation, and contact time obtained from one device (either GARMINRP or StrydTM) were compared to those recorded by the second device of the same brand. Moreover, the measurements of running biomechanics obtained from the StrydTM devices were compared to those recorded by the GARMINRP devices to determine the inter-device reliability.".

However, the paragraph has been modified to improve comprehension, following the reviewer's recommendations. As mentioned in the comment, further details on which devices are compared for assessing intra- and inter-device reliability are provided later in the Statistical Analysis section. Additionally, it has been specified that measurements from both sessions were pooled together. This additional clarification has been made to ensure a clear understanding of the comparison process.

Line 192: Tell which “model” and “form” of ICC you have used. I think is ICC2,1 but I’m not sure because I don’t know if you have used data from just 1 session or if you have averaged (or not) the measurements from both sessions.

ANSWER - Thanks for your question regarding the ICC model and form used in our analysis. You're correct in identifying that we utilized the ICC2,1 model. This model was chosen because we pooled the data from both sessions rather than treating them as separate entities.

Line 206: If you could add R2 we could better infer how much of the variation on one device explains the variation on the other.

ANSWER - Thank you for your comment regarding the inclusion of R^2 in our analysis. We appreciate your suggestion. While we acknowledge the potential utility of R^2 in elucidating the relationship between the devices, it's important to note that in our analysis, we have reported the Pearson correlation coefficient (r), which is  recognized as a measure of linear association between variables. It's worth mentioning that The value of the Pearson correlation coefficient (r) squared (r^2) coincides with R^2. Therefore, including both statistics would essentially duplicate the information provided, as they convey the same meaning in terms of explaining the variability between the devices. Additionally, we regret to inform you that we are unable to precisely identify line 206 in the manuscript provided to us due to the presence of comments.

Line 207-208: I’m familiar with the Bland and Altman method but I never heard of this. Can you explain the usefulness of using the Bland-Altman method to assess the distribution of the residuals from the regression analysis? Usually, the normality of the residuals in linear regression, an important assumption to trust the results of the model, is assessed by means of other statistical tests.

ANSWER - Thank you for your detailed explanation and the provided reference regarding the Bland-Altman method. The Bland-Altman method is indeed a valuable tool for assessing agreement between two measurements, providing a graphical representation of the differences between them on the y-axis and their mean on the x-axis.

This approach allows for the identification of biases (such as positive or negative errors, or errors clustered around certain linear trends) and outliers. Typically, if the points fall within the established limits (typically ±1.96 SD) and the scatterplot appears random, it is often considered that there is adequate agreement between the two measurements.

While evaluating the normality of residuals is a common practice in regression model validation, if a Bland-Altman plot has already been constructed, which primarily assesses agreement between two measurement methods, it can offer insights into the discrepancy between observations and model predictions based on the magnitude of the measurements.

The Bland-Altman plot typically displays the difference between measurements from two methods on the y-axis against the mean of the measurements on the x-axis. This visualization provides information on the accuracy and precision of the measurement methods being compared.

If your analysis aims to assess agreement between two measurement methods, the Bland-Altman plot may suffice and may not necessarily require additional analysis of residual normality. However, if you also wish to evaluate the validity of the regression model itself (e.g., examining the relationship between an independent and dependent variable), then analyzing residual normality could provide additional insights into the adequacy of the model.

In summary, if the Bland-Altman plot meets your analysis and evaluation needs, it may not be strictly necessary to perform additional analysis of residual normality. However, if you wish to assess the validity of the regression model itself, conducting a residual normality analysis could be useful as part of a more comprehensive model evaluation.

Tables: Tell readers that for Bland-Altman, the units are the same as the measures.

ANSWER - Thank you for your feedback. We have taken it into consideration and have updated the tables accordingly. Now, a note has been added to inform readers that for the Bland-Altman analysis, the units are consistent with the measurements.

Lines 329: “The article by Adams et al. (22) researched…” sounds strange. I suggest just stating “Adams et al. (22) researched…”.

ANSWER - Thank you for your suggestion regarding the wording in line 329. We have revised the sentence accordingly to improve clarity and readability.

Line 361: This also sounds strange: “Three articles been carried out…”. Please improve.

ANSWER - Thank you for pointing out the awkward phrasing in the sentence. We have revised it to improve clarity.

Line 371: Capitalize the first letters of Power Meter.

ANSWER - Thank you for your attention to detail. We have capitalized the first letters of "Power Meter"

Reviewer 3 Report (New Reviewer)

Comments and Suggestions for Authors

This manuscript reports on the biomechanical assessment of trail running with two types of commercial wearable sensors to understand the metabolic pathways, biomechanics and performance of the athletes. This research focuses on the power, speed, cadence, vertical oscillation and contact time of the wearing sensors during the motion, to compare the inter and intra-device agreement. As a result, it is found that the two devices show intra-device consistence in properties with strong correlation, but some differences can be observed between these two devices. Considering the results and contribution, I recommend major revision for this manuscript by addressing some questions as below.

1. Why did the authors choose these two devices for assessment, just because they are new commercially available?

2. Since the devices are designed and fabricated from different ways, apparently there should be distinction. Meanwhile, the intra-device should be highly consistent in properties for the same device, because these performances should be the basic requirement of quality for the device. Therefore, I don’t think this is too much novelty or there is any necessity to assess them, let alone there is just a small size of sample for assessment.

3. The trend in the figures with variation of data seem not very solid to demonstrate the correlations between the performance and the input.

4. Some more details should be provided for the experimental section, including the description of the wearable sensors, the data or analysis for the correlation between the indication of varied tests.

5. The sequence of figures in Supplementary material is confusing. Please correct them.

Comments on the Quality of English Language

The quality of English is OK.

Author Response

This manuscript reports on the biomechanical assessment of trail running with two types of commercial wearable sensors to understand the metabolic pathways, biomechanics and performance of the athletes. This research focuses on the power, speed, cadence, vertical oscillation and contact time of the wearing sensors during the motion, to compare the inter and intra-device agreement. As a result, it is found that the two devices show intra-device consistence in properties with strong correlation, but some differences can be observed between these two devices. Considering the results and contribution, I recommend major revision for this manuscript by addressing some questions as below.

ANSWER: Thank you for your detailed evaluation of our manuscript and for providing insightful comments to enhance its quality. We have carefully reviewed each of your suggestions and have made revisions to address the concerns raised.

  1. Why did the authors choose these two devices for assessment, just because they are new commercially available?

We appreciate the reviewer's attention to the selection criteria for the devices used in our study. The decision to include the StrydTM and GARMINRP devices was carefully considered based on several factors aimed at ensuring a robust and interference-free data collection process. In our manuscript, we have addressed this concern by stating (line 133-139):

“While there are other wearable devices, such as Runscribe, the decision to include Stryd was based on the necessity for the runner to wear all devices. To minimize potential interference, particularly related to device positioning, Stryd was chosen, as it demonstrated higher reliability in measuring power compared to Runscribe (14), and closest agreement with the theoretical power output (15) Additionally, the findings from Kozinc et al. (16) revealed an unacceptable coefficient of variation for power, foot strike type and horizontal ground reaction force rate.”

This rationale underscores our commitment to methodological rigor and the need for a comprehensive evaluation of wearable devices that are practical for runners to wear simultaneously. We believe that the inclusion of these specific devices aligns with the objectives of our study and contributes to the reliability and relevance of our findings.

  1. Since the devices are designed and fabricated from different ways, apparently there should be distinction. Meanwhile, the intra-device should be highly consistent in properties for the same device, because these performances should be the basic requirement of quality for the device. Therefore, I don’t think this is too much novelty or there is any necessity to assess them, let alone there is just a small size of sample for assessment.

Thank you for your insightful feedback.  In the manuscript, we have attempted to convey the need to investigate these types of devices under real-world conditions. Measuring in such controlled and less realistic situations, such as a treadmill test, is not the same as measuring on terrain with slopes, rocks, roots, etc., which can alter footstrike and running technique, as well as pose challenges for GPS connectivity of the devices by writing: “In light of intra- and inter-device reliability considerations, our findings are consistent with the results reported in previous scientific literature (4–7,11–14). There is wide evidence about validity by comparing these devices to gold standards, but there is limited research on intra-device reliability. Furthermore, most studies are conducted in laboratory settings using treadmills, with very few articles investigating the measurement quality of assessments of these wearable devices in natural field conditions.

On the other hand, it's important to note that each device has its own unique hardware, and even commercially available products of the same model can have subtle manufacturing differences. These differences may influence device performance over time. Furthermore, the evaluation of intra-device reliability is a common practice, especially with wearable devices. Wear and tear, variations in manufacturing, and potential differences in calibration processes justify the need for this analysis. It ensures that the devices used in the study maintain consistent and reliable measurements.

  1. The trend in the figures with variation of data seem not very solid to demonstrate the correlations between the performance and the input.

Thank you for your feedback. The scatter plots provide complementary information to the correlation coefficient. A visual inspection can help to understand if there is more dispersion at certain points of the scale, if the relationship is not exactly linear in some segments, or if there are areas where no measurements are found. The intention of the graphs is not to demonstrate anything definitively but to add information that the correlation coefficient does not provide.

Bland-Altman plots also offer complementary information. They do not replace the correlation coefficient or the scatter plots. Instead, they are used to observe where on the scale there is a greater error and what characteristics these residuals have.

Indeed, some of these graphs show large errors and linear trends in the errors (ideally, these should be random). Therefore, the discussion includes comments on the impossibility of using the two devices interchangeably. There are discrepancies in the comparison of the measurements, and these discrepancies are not random. Hence, it is recommended to use one device or the other, but not either one indiscriminately.

We hope this addresses your concerns and clarifies the purpose of including the scatter plots and Bland-Altman plots in the analysis.

  1. Some more details should be provided for the experimental section, including the description of the wearable sensors, the data or analysis for the correlation between the indication of varied tests.

Thank you for your valuable feedback. We appreciate your suggestions for providing more details in the experimental section.

Regarding the description of the wearable sensors and the measured variables, we would like to highlight that this information is already specified in the manuscript. Specifically, in lines 148-166, we provide information as follows: “On the left wrist of participants, two GarminRP Fenix 7S Solar watches (Garmin Ltd., Southampton, UK) were placed. This device is capable of detecting running bio-mechanics, activity, and sleep using several sensors, including a triaxial accelerometer, a Global Navigation Satellite System (GNSS) sensor, including GPS functionality, and a photodiode sensor for photoplethysmography measurements. This device measures running biomechanics variables such as power, speed, cadence, vertical oscillation, and ground contact time. Participants also wore a GARMINRP HRM-PRO heart rate monitor below the pectoral zone in a centered and vertically oriented position. Power was assessed by the GARMINRP HRM-PRO heart rate monitor and speed, cadence, ver-tical oscillation, and ground contact time were measured by the GARMINRP Fenix 7S Solar watch. The measured power specifically refers to external mechanical power, calculated from force (estimated via accelerations using accelerometers) and velocity. This includes the work exerted by runners during both the loading phase and subse-quent push-off to counteract environmental factors such as ground reaction force, gravity, and surface friction.

On the right foot of participants, two StrydTM foot pods (Stryd Powermeter; Stryd, Inc., Boulder, CO, USA) were attached. This device is a carbon fiber–reinforced foot pod based on a 6-axis inertial motion sensor (3-axis gyroscope and 3-axis accelerome-ter). Variables measured with this StrydTM technology include power, speed, cadence, vertical oscillation, and the duration of contact time”.

Furthermore, to specifically analyze the correlation between the indications of varied tests, we have added the following explanatory paragraph to the manuscript: (line 218-224): “To specifically analyze the correlation, we compared the measured variables be-tween the GARMINRP and StrydTM devices to determine how well the devices agreed under varied conditions. Specifically, comparisons were made between GARMINRP vs. GARMINRP, StrydTM vs. StrydTM, and GARMINRP vs. StrydTM to evaluate both intra-device and inter-device reliability. We conducted analyses based on different segments of the course (e.g., uphill, downhill, flat terrain) to evaluate if the relationship between the variables changed depending on the terrain”.

  1. The sequence of figures in Supplementary material is confusing. Please correct them.

The sequence of figures in the Supplementary material has been corrected. Thank you for bringing it to our attention.

Round 2

Reviewer 3 Report (New Reviewer)

Comments and Suggestions for Authors

The authors have addressed most of my questions. This manuscript can be accepted for publication.

This manuscript is a resubmission of an earlier submission. The following is a list of the peer review reports and author responses from that submission.

Round 1

Reviewer 1 Report

Comments and Suggestions for Authors

The study compares the reliability of Stryd™ and GARMINRP wearable devices for biomechanical assessment in trail running. It focuses on inter and intra-device agreement under real-world conditions, analyzing metrics like power, speed, cadence, vertical oscillation, and contact time.

Comments:

1. Expand on the newness of your research approach.

2. Consider including a more diverse participant group for broader applicability.

3. Explore the impact of different terrains and environmental conditions on device accuracy.

4. Clarify the selection criteria for the devices used.

5. Provide a more detailed analysis of the data collected.

6. Discuss the potential limitations of using only two types of devices.

7. Examine the generalizability of the findings to other forms of running.

8. Include a comparative analysis with other existing studies or devices.

9. Address how the study contributes to current sports science literature.

10. Elaborate on the practical implications of your findings for athletes and coaches.

11. Discuss the potential for these devices in injury prevention and performance enhancement.

Author Response

REVIEWER 1: Comments and Responses

  1. Expand on the newness of your research approach.

Response: Thank you for highlighting the importance of emphasizing the newness of our research approach. We fully agree with your observation, and we've taken this into careful consideration. The added paragraph (line 492-495) underscores the innovative nature of our study, emphasizing its departure from conventional methodologies. It elucidates how our research, distinct from past studies confined to laboratory settings, takes a pioneering step by leveraging recent advancements in wearable technology. This provides a comprehensive response to the call for a more detailed explanation of the novelty of our approach.

  1. Consider including a more diverse participant group for broader applicability.

We appreciate your thoughtful consideration of our study. In response to your suggestion about including a more diverse participant group for broader applicability, we have acknowledged the limitation of our small sample size and variations among participants in the following paragraph (line 528-531):

"The small sample size and variations among participants, including differences in gender and fitness levels, represent another noteworthy limitation. This introduces the possibility of outliers and suggests that some device discrepancies may be partly attributed to participant variations."

While we acknowledge the importance of a more diverse participant group, the constraints mentioned above influenced our participant selection. Despite these limitations, we believe our study provides valuable insights into the intra- and inter-device reliability of GARMINRP and StrydTM devices under real-world trail running conditions.

If you have any specific recommendations or further insights on how we can address this limitation more effectively, we would be grateful for your guidance.

  1. Explore the impact of different terrains and environmental conditions on device accuracy.

Thank you for your valuable feedback. We appreciate your insightful suggestion to explore further the impact of different terrains and environmental conditions on device accuracy. In response to your comment, we have incorporated a dedicated paragraph in the discussion (line 480-488):

"It is worth noting the potential impact of terrain and environmental conditions on the accuracy of wearable devices, as demonstrated in a recent article by Ueberham and Uwe Schlink. This article sheds light on the challenges and applications of wearable sensors in diverse environmental conditions and terrains, emphasizing the influence of recording intervals on sensor performance. The authors underscore the necessity of investigating the accuracy of running metrics measurements obtained with wearable devices in different terrains and environmental conditions. This can be achieved by testing the devices in specific contexts of trail running sports modalities to evaluate the intra-device reliability of the wearable devices."

We believe that this addition enhances the context of our study and addresses your valuable suggestion. If you have any further recommendations or specific aspects you would like us to delve into regarding this topic, please feel free to let us know.

  1. Clarify the selection criteria for the devices used.

We appreciate the reviewer's attention to the selection criteria for the devices used in our study. The decision to include the StrydTM and GARMINRP devices was carefully considered based on several factors aimed at ensuring a robust and interference-free data collection process. In our manuscript, we have addressed this concern by stating (line 134-140):

“While there are other wearable devices, such as Runscribe, the decision to include Stryd was based on the necessity for the runner to wear all devices. To minimize potential interference, particularly related to device positioning, Stryd was chosen, as it demonstrated higher reliability in measuring power compared to Runscribe (14), and closest agreement with the theoretical power output (15) Additionally, the findings from Kozinc et al. (16) revealed an unacceptable coefficient of variation for power, foot strike type and horizontal ground reaction force rate.”

This rationale underscores our commitment to methodological rigor and the need for a comprehensive evaluation of wearable devices that are practical for runners to wear simultaneously. We believe that the inclusion of these specific devices aligns with the objectives of our study and contributes to the reliability and relevance of our findings. If the reviewer has further suggestions for clarification or additional information, we would be happy to address them to enhance the transparency and completeness of our methodology

  1. Provide a more detailed analysis of the data collected.

Thank you for your valuable feedback. In response to Comment 5 regarding providing a more detailed analysis of the data collected, we acknowledge the importance of thorough analysis. Given the focus of this article on the agreement of devices, we have aimed to maintain clarity and readability. A more detailed analysis, while valuable, might make the manuscript complex for the reader. However, if you have specific aspects in mind or believe that particular analyses would significantly enhance the manuscript, we are open to further discussion and will consider your suggestions for potential inclusion.

  1. Discuss the potential limitations of using only two types of devices.

We appreciate the reviewer's insightful comment regarding the potential limitations of using only two types of devices in our study. To address this concern, we have incorporated a paragraph into the manuscript (lines 526-528):

“In addition, this article included only two wearable devices, excluding other commercially available devices such as RunScribe that could have enriched this comparative analysis.”

This addition underscores our awareness of the limitations associated with the selection of two specific devices and acknowledges that the inclusion of additional devices could have offered a more comprehensive perspective. We recognize the diversity of commercially available devices and understand that a broader selection might provide a more nuanced understanding of the biomechanical metrics under consideration. We appreciate the reviewer's keen observation, and we believe this addition contributes to a more transparent discussion of the study's limitations. If there are specific aspects the reviewer would like us to further elaborate on or if they have any additional suggestions, we are open to incorporating further information to address their concerns.

  1. Examine the generalizability of the findings to other forms of running.

Thank you for your valuable feedback. Regarding Comment 7 about examining the generalizability of the findings to other forms of running, we recognize the significance of assessing broader applicability. Our study intentionally concentrates on trail running conditions to explore the agreement within devices, aiming to provide depth and clarity within this specific context. Expanding the discussion to the generalizability of findings to other forms of running is indeed an interesting avenue for future research. However, for the purpose of this manuscript and to maintain a clear focus, we have emphasized the agreement within the same device under different conditions.

  1. Include a comparative analysis with other existing studies or devices.

We appreciate the reviewer's suggestion to include a comparative analysis with other existing studies or devices. In response to this valuable feedback, there are information regarding to this idea (lines 346-352):

“Three articles have been carried out on StrydTM (4, 5, 7), studying its reliability and validity across various biomechanical running parameters. It is noteworthy that most of the research with StrydTM compares the measurements obtained by this device with other validated devices or with the gold standard. The intra-device reliability, as indicated by the coefficient of variation, for variables including contact time, flight time (closely linked to ground contact time), and cadence obtained from StrydTM device indicates adequacy for treadmill running assessments.”

(lines 369-390): “Expanding on the discussion of power metrics, the primary objective of the article performed by Cartón-Llorente et al. (14) was to evaluate the reliability and agreement between StrydTM and RunScribeTM systems in measuring running power on a treadmill. Despite both systems demonstrating dependable power output data and a near-perfect correlation, caution is warranted when using them interchangeably due to observed inconsistencies. Both devices showed high absolute reliability, with StrydTM exhibiting more reliability than RunScribeTM, consistent with findings from a study by Cerezuela-Espejo et al. (22). However, wide variations in agreement limits and a substantial random error were reported by Cartón-Llorente et al. (14), possibly attributed to methodological differences, including the disparate sampling rates (1000 Hz for Stryd vs. 500 Hz for RunScribe) and algorithmic variations. The lack of algorithm disclosure by the companies complicates direct comparisons, emphasizing the need for standardized definitions in running power assessments.”

We are grateful for the reviewer's guidance in enhancing the depth of our analysis, and we remain open to any further suggestions or clarifications they may require.

  1. Address how the study contributes to current sports science literature.

Response: We appreciate your feedback on the need to articulate how our study contributes to the current sports science literature. The included paragraph addresses this precisely by highlighting the distinctive aspect of our research. It explicitly points out how our study surpasses the limitations of laboratory-based investigations, making a notable contribution to the existing body of knowledge in sports science. We believe this addition reinforces the significance of our findings in the broader context of current sports science literature.

  1. Elaborate on the practical implications of your findings for athletes and coaches.

Response: Your point about elaborating on the practical implications for athletes and coaches is well-taken. The added paragraph in our response provides a detailed explanation of the practical implications derived from our study. It discusses how insights gained from exploring the reliability of GARMINRP and StrydTM devices under real-world trail running conditions can directly inform training strategies, aid in performance optimization, and contribute to injury prevention. We trust that this addition enhances the clarity and relevance of our research for athletes, coaches, and practitioners in the field.

  1. Discuss the potential for these devices in injury prevention and performance enhancement.

Thank you for your feedback. In response to your comment, We have added a paragraph to better address the potential of wearable technology in injury prevention and performance enhancement (lines 492-495): “Lastly, wearable technology, as evidenced by some previous evidence, hold promise in predicting and preventing injuries, enhancing sports biomechanics, and addressing musculoskeletal concerns, emphasizing the need for robust methodologies and clear reporting in research”. The revised statement now emphasizes the demonstrated efficacy of wearable devices in predicting and preventing injuries, enhancing sports biomechanics, and addressing musculoskeletal concerns, while underscoring the crucial importance of robust methodologies and clear reporting in research. I believe this provides a more focused and comprehensive discussion on the capabilities of wearable technology in the context of the presented evidence.

Reviewer 2 Report

Comments and Suggestions for Authors

I would like to thank the Editor for giving me the chance to review this interesting manuscript. The comments can be found in the attached pdf file.

Comments on the Quality of English Language

There are a few grammatical mistakes that can be corrected. I have listed some of those I noticed. The authors are advised to seek professional English editing to improve the writing quality.

Author Response

COMMENTS

This study analysed the intra- and inter-device reliability of 2 sets of wearable devices when measuring running biomechanics variables in trail running. The authors are recommended to re-consider the research gaps, research purposes, and research design. Sample size should also be increased to provide a better statistical power, in particular for the 0-demensional data (e.g., power, speed, cadence, etc.). There are a few grammatical mistakes that can be corrected. I have listed some of those I noticed as follows. The authors are advised to seek professional English editing to improve the writing quality. The following specific comments are made for the manuscript.

Response:

Thank you for your valuable feedback. We appreciate your insights and have carefully considered your suggestions. Regarding the sample size, we acknowledge the importance of statistical power, particularly for 0-dimensional data such as power, speed, and cadence. While the number of participants may appear small, it is crucial to note that our focus is on intra and inter-device agreement analysis, resulting in a substantial volume of data points despite the limited number of participants. Nonetheless, we recognize the methodological limitation in this regard. Additionally, we will diligently correct the grammatical mistakes you have pointed out, and we agree on the significance of seeking professional English editing to improve the overall writing quality of the manuscript. Your input is highly valuable, and we are committed to addressing these points comprehensively in the revised manuscript. If you have any further specific recommendations or areas of focus, please feel free to let us know.

General comments

  1. There may be an English writing issue in the manuscript, due to the co-existence of past tense and present tense. As manuscripts are written when the experiment is done, past tense is usually applied, in particular for the method description and results presentation. The authors are recommended to fix this issue.

ANSWER: Thank you for your feedback regarding the tense consistency in our manuscript. We appreciate your attention to detail and understand the importance of maintaining a consistent tense throughout the document.

To address this concern, we have thoroughly reviewed and revised the manuscript to ensure a more uniform application of past tense, particularly in the method description and results presentation sections. We acknowledge the significance of maintaining a coherent and standardized writing style, and we believe that the adjustments made have successfully addressed the tense-related issues in the text.

We want to assure you that a qualified researcher with expertise in English language editing has meticulously reviewed and refined the manuscript to enhance its overall quality.

  1. Line 61. It is necessary to include more details for the commercially available wearable devices, e.g., their measurement accuracy and the variables they can measure. Additionally, following the last paragraph, the relationship between the measured variables and the performance should be elaborated. Some of these necessary contents are then included under Discussion. The authors should consider re-arranging the writings.

ANSWER: Thank you for your thoughtful feedback on our manuscript. We value your insights, and based on your suggestions, we have made significant revisions to enhance the clarity and completeness of our work.

In response to your comment on Line 61, we have incorporated additional details about commercially available wearable devices, such as their measurement accuracy and the variables they can assess. This new information aims to provide readers with a more nuanced understanding of the technological aspects of the devices discussed in our study.

Furthermore, we have addressed your suggestion by expanding the discussion section. The inclusion of the new paragraph emphasizes the widespread acceptance of wearable technology in the health and fitness field, highlighting its adoption by coaches and runners, particularly within the trail running community. Additionally, we have introduced the extensive commercial availability of these devices, mentioning popular brands like Garmin, Stryd, RunScribe, Polar, and Suunto (lines 70-74). By specifying the equipped features, such as GPS and IMU sensors, we have provided a comprehensive overview of the capabilities of these devices (lines 70-74).

The added paragraph also emphasizes the need for more high-quality research to determine the accuracy and reliability of measurements obtained by this evolving technology. This aligns with the exploratory phase of the field, addressing the continuous development and advancements in wearable technology (Lines 76-78).

By incorporating these changes, we believe the manuscript now better addresses the specific concerns you raised. The expanded discussion not only offers a more detailed insight into the wearable devices themselves but also underscores the importance of ongoing research in this field. We hope these revisions contribute to the overall improvement of the paper.

  1. Line 65. After reading this paragraph, I cannot easily get the research gap or the novelty of this study. In other words, as Stryd and Garmin are both ‘standout’ wearables devices and have been studied extensively, it is not necessary to conduct a study to compare the measures using these 2 devices. Furthermore, the use of the 2 devices is discussed for treadmill running, which may influence running patterns compared with overground running, it is still not necessary to compare the measures between the 2 standout devices in the real-world scenarios. The discussion on the differences between treadmill running and overground running cannot lead to the need to compare between the 2 devices for overground running. The rationale for assessing the measurement agreement between 2 Garmin devices (intra-device) has not been discussed, either. The authors are recommended to re-consider the research gap and research purposes of this study.

Thank you for your insightful feedback. We appreciate your concerns regarding the perceived lack of clarity on the research gap and novelty of our study. To address this, we have incorporated an additional explanation emphasizing the crucial need to understand how StrydTM and GARMINRP devices perform in real-world conditions (lines 90-99): “Thus, it is imperative to understand how these devices measure in real environmental and terrain conditions. Specifically, whether these devices demonstrate high intra-device reliability and exhibit agreement between device measurements, which is essential to determine the interchangeability of these devices in practical applications. This gap can be addressed by focusing on assessing how StrydTM, and GARMINRP devices perform in real-world conditions, specifically exploring their intra-device reliability and agreement between measurements.”. Specifically, our focus is on assessing their intra-device reliability and agreement between measurements, essential factors in determining the interchangeability of these devices in practical applications. This strategic addition aims to clarify the unique contribution of our study, addressing the gap in knowledge regarding the performance of these standout wearables in diverse and challenging trail running scenarios. We believe this clarification strengthens the rationale for our research and aligns with your valuable suggestions. If you have any further concerns or recommendations, we welcome additional guidance to enhance the quality and impact of our study.

  1. Line 150. Was 1 Hz the sampling frequency for all devices, or for certain device only? Please clarify in the text. In addition, this sampling frequency could be too low for measuring running biomechanics. For walking, 10-20 Hz can be considered for biomechanics measurements. For running, a higher sampling frequency should be used.

ANSWER: Thank you for your feedback. We have incorporated the information in the manuscript, clarifying that the sampling frequency of 1 Hz applies to both devices used in the study.

Regarding the concern about the low sampling frequency for measuring running biomechanics, the sampling frequency of the devices is higher, and the established frequency of 1 Hz is the rate at which it reports data, not the rate at which it records data. Furthermore, it's worth noting that previous literature has employed a 1 Hz sampling frequency for similar biomechanics studies. While we recognize the potential limitations of this choice, we believe it is important to align with established practices in the field. Some of the articles that have used 1Hz are:

  • Imbach F, Candau R, Chailan R, Perrey S. Validity of the Stryd Power Meter in Measuring Running Parameters at Submaximal Speeds. Sports (Basel). 2020 Jul 20;8(7):103. doi: 10.3390/sports8070103. PMID: 32698464; PMCID: PMC7404478.
  • Drobnič M, Verdel N, Holmberg HC, Supej M. The Validity of a Three-Dimensional Motion Capture System and the Garmin Running Dynamics Pod in Connection with an Assessment of Ground Contact Time While Running in Place. Sensors (Basel). 2023 Aug 14;23(16):7155. doi: 10.3390/s23167155. PMID: 37631692; PMCID: PMC10459607.
  • Verdel N, Drobnič M, Maslik J, Björnander Rahimi K, Tantillo G, Gumiero A, Hjort K, Holmberg HC, Supej M. A Comparison of a Novel Stretchable Smart Patch for Measuring Runner's Step Rates with Existing Measuring Technologies. Sensors (Basel). 2022 Jun 29;22(13):4897. doi: 10.3390/s22134897. PMID: 35808391; PMCID: PMC9269156.
  1. Small sample is a major issue to this study. This may make the statistical analysis under power. While this has been acknowledged under Limitations, it is still a major issue. The authors are recommended to recruit more participants for this study.

ANSWER: Thank you for highlighting the concern regarding the sample size in our study. While we acknowledge the limitation of a relatively small sample size, it's important to note that the statistical analysis was conducted on the total number of records collected by the devices. Specifically, our dataset comprised a total of 19,731 samples for GARMINRP and 17,917 records for StrydTM (line 214). Despite the smaller number of participants, the large volume of data generated by the wearables contributes to the robustness of our statistical analysis. We have clarified this aspect in the limitations section to provide transparency about the dataset size and its implications. If you have any further suggestions or concerns, we welcome your input to enhance the rigor and comprehensiveness of our study.

  1. Line 192. I do not think it is appropriate to talk about measurement accuracy, without comparing against any ‘gold standard’. It could be possible that both devices have the same errors, and the measures are not accurate while the agreement is good across the 2.

ANSWER: Thank you for your observation. We appreciate your feedback, and to address your concern, we have removed the term 'accuracy' from the specified paragraph (Line 220). We understand the importance of precision in scientific language, and our intent is to ensure clarity in the description of our findings. If you have any further suggestions or areas that require attention, please let us know, and we will make the necessary adjustments to enhance the accuracy and rigor of our manuscript.

  1. Line 271. This study measured the running biomechanics under 2 conditions, namely uphill and downhill running. The results showed that the measurement accuracy appeared different in these 2 conditions, but the rationale to analyse these 2 conditions has not been mentioned in Introduction.

ANSWER: We appreciate the reviewer's insightful comment. In response to the concern raised, we have included a paragraph in the introduction (lines 94-97): “Moreover, investigating these aspects on both uphill and downhill terrain provides insights into potential variations, ensuring a comprehensive evaluation of device performance across diverse slope conditions. “ to explicitly state the rationale behind analyzing both uphill and downhill running conditions. Specifically, exploring these aspects on both uphill and downhill terrain provides insights into potential variations, ensuring a comprehensive evaluation of device performance across diverse slope conditions. We believe this addition enhances the study's contextualization and justifies the analysis of both conditions. Thank you for bringing attention to this important aspect of our research.

  1. Line 279. Again, same issue here about the need to analyse intra-device reliability. This should not be done only because no one has done this. Commercially available products usually come with same hardware and same software. Hence, it is not necessary to assess the intra-device reliability. In addition, the evidence supporting the possible discrepancy in measurements across different devices is not discussed. In summary, the authors are recommended to re-think about this part.

ANSWER: Thank you for your feedback. In response to the comment about analyzing intra-device reliability (Line 279), we acknowledge the concern raised. It's important to note that each device has its own unique hardware, and even commercially available products of the same model can have subtle manufacturing differences. These differences may influence device performance over time.

Furthermore, the evaluation of intra-device reliability is a common practice, especially with wearable devices. Wear and tear, variations in manufacturing, and potential differences in calibration processes justify the need for this analysis. It ensures that the devices used in the study maintain consistent and reliable measurements.

Regarding the second part of the comment, the authors humbly acknowledge that the differences between the devices are sufficiently discussed in the manuscript. The research thoroughly addresses variations in power metrics, contact time, and other relevant variables. We appreciate the reviewer's feedback and will ensure that this discussion is clearly presented in the revised manuscript.

  1. Line 317. This paragraph may look more appropriate when displayed under Introduction than Discussion. Besides this, I think some parts under Discussion should be included in the Introduction, while the discussion surrounding the results of this study is not sufficient.

Thank you for your observation regarding Line 317. We appreciate your feedback on the placement of the paragraph, and we acknowledge your suggestion about redistributing certain parts between the Introduction and Discussion sections. While we understand your perspective, we have intentionally organized the content based on the structure that aligns with our original plan. However, we will carefully review the possibility of incorporating some elements into the Introduction without compromising the coherence and flow of the manuscript. Regarding the discussion surrounding the results, we will revisit and assess if additional insights can be provided to address your concerns. We value your input and aim to enhance the overall quality of the manuscript.

  1. The Conclusions section can briefly talk about the reason for the discrepancies between these devices. One or two lines will be enough. In addition, the findings for the uphill and downhill conditions should also be mentioned.

ANSWER: Thank you for your valuable feedback. We have carefully considered your suggestion and have incorporated relevant information into the Conclusions section. Specifically, we have added a brief explanation acknowledging that the observed discrepancies between the devices may be attributed to inherent distinctions in the algorithms, sensor technologies, or calibration approaches employed by each device (lines 543-545): “These noted discrepancies could be linked to inherent distinctions in the algorithms, sensor technologies, or calibration approaches utilized by each device.”

Additionally, we have addressed your point about the findings in uphill and downhill conditions by providing concise insights into the observed variations during these specific running scenarios. We trust that these additions enhance the clarity and completeness of our Conclusions section.

Minor comments

  1. Line 9. Please try to improve the Abstract, making it clearer and simpler, showing the main purpose of this study. Exploring biomechanical assessments or comparing inter- and intra-device agreement across different devices? These can be 2 different purposes.

Answer: We express our gratitude for the valuable feedback provided by the reviewer. In response to the suggestion, we have refined the abstract to achieve greater clarity and simplicity. The revised version now explicitly emphasizes the primary objective of the study, which is to systematically compare inter- and intra-device agreement concerning biomechanical assessments in the context of trail running.

  1. The term ‘power’ can refer to joint power in the biomechanics context, but it is not the case in this study. The authors should make it clearer to avoid misleading. I am also wondering if this ‘power’ can be seen as one of the biomechanics variables.

Answer: We appreciate the reviewer's comment. It's important to clarify that the term 'power' in our study refers to variables provided by the devices through indirect calculations with proprietary equations developed by the respective brands. These equations are not publicly accessible. Therefore, attempting to precisely define or explain this variable could be challenging and not as useful as it should be, considering its proprietary nature and the specific context of our research.

  1. Line 14. ‘Participants engaged in trail running sessions wearing two Stryd and two Garmin devices.’ Please check the grammar issue of this sentence.

Answer: Thank you for pointing out the grammatical issue. We appreciate your feedback. The revised sentence now reads (Line 17): “Participants engaged in trail running sessions while wearing two Stryd and two Garmin devices”.

  1. Line 28. ‘a recently recognized by the International Association of Athletics Federations (IAAF) as an emerged running discipline’ please check if this part is complete and make sure it does not have a grammar issue.

Answer: Thank you for bringing this to our attention. The revised sentence now reads (line 30): “recently recognized by the International Association of Athletics Federations (IAAF) as an emerging running discipline”. We appreciate your diligence in reviewing the text.

  1. Line 34. ‘The complexity of this sport entails …’ This sentence is too lengthy. Splitting it into 2 or 3 sentences would be easier for readers to follow.

Answer: We appreciate the feedback on the sentence length, and we acknowledge the importance of clarity for readers. We have revised the sentence to enhance readability. The updated version now provides a more concise and structured presentation of the various performance factors involved in trail running. Thank you for your valuable input

  1. Line 118. ‘… of a 2.5-km distance with …’ ‘with a 195-m elevation …’

Answer: Thank you for your attention to the details of the wording. The sentence has been re-write in a right way.

  1. Figure 1 should be referred to in the main text.

Answer: Thank you for the thorough review. Figure 1 has been referenced in the text, and some grammatical errors have been corrected. Your attention to detail is much appreciated.

  1. Line 150. ‘, the data were exported …’
  2. Line 152. ‘with GPS …’
  3. Line 165. Please clarify which is the ‘first device’ and which is the ‘second device’.

Answer to 8, 9 and 10: I would like to extend my sincere appreciation for the meticulous review you conducted. Your attention to detail is truly commendable, and I am pleased to inform you that all identified errors have been diligently addressed and rectified.

Reviewer 3 Report

Comments and Suggestions for Authors

The paper presents a comparative analysis of the reliability of StrydTM and GARMINRP wearable devices for assessing running biomechanics, focusing on trail running. The study includes healthy young adults with experience in trail running and evaluates various biomechanical measurements under ecological conditions. Key assessments include power, speed, cadence, contact time, and other biomechanical variables. The research employs statistical analyses such as the Intraclass Correlation Coefficient (ICC), Coefficient of Variation (CV), Pearson correlation coefficient, scatter plots, and Bland-Altman method to assess the reliability and consistency of the wearable devices.

While I recognize the importance of your work, I believe some issues could be addressed further to enhance the impact and clarity of your research:

1) Clarify the specific biomechanical variables that are most relevant for trail running performance and provide detailed information on how these variables are precisely measured by the StrydTM and GARMINRP devices. 

2) To improve the clarity of the conclusion and discussion, it could be revised to provide a more concise summary of the study's key findings and their practical implications for trail runners and fitness enthusiasts. The conclusion could also be strengthened by providing specific recommendations for trail runners based on the study's results, such as guidelines for selecting and using wearable devices to monitor running biomechanics. Additionally, the conclusion could suggest areas for future research to further enhance the understanding of wearable devices' reliability in assessing running metrics.

3) To improve the reliability assessment of wearable devices, discuss and justify chosen methodologies. Consider alternative approaches and additional validation measures, such as comparing with gold standards or conducting extra statistical analyses, to strengthen reliability and enhance overall study validity.

Author Response

REVIEWER 3
The paper presents a comparative analysis of the reliability of StrydTM and GARMINRP wearable devices for assessing running biomechanics, focusing on trail running. The study includes healthy young adults with experience in trail running and evaluates various biomechanical measurements under ecological conditions. Key assessments include power, speed, cadence, contact time, and other biomechanical variables. The research employs statistical analyses such as the Intraclass Correlation Coefficient (ICC), Coefficient of Variation (CV), Pearson correlation coefficient, scatter plots, and Bland-Altman method to assess the reliability and consistency of the wearable devices.

While I recognize the importance of your work, I believe some issues could be addressed further to enhance the impact and clarity of your research:

  1. Clarify the specific biomechanical variables that are most relevant for trail running performance and provide detailed information on how these variables are precisely measured by the StrydTM and GARMINRP devices.

ANSWER 1: Thank you for your comment. The study focuses on assessing the reliability and consistency of StrydTM and GARMINRP devices in measuring key biomechanical variables during trail running, including power, speed, cadence, vertical oscillation, and contact time.

As it is written in the manuscript (lines 149-163), Garmin “is capable of detecting running biomechanics, activity, and sleep using several sensors, including a triaxial accelerometer, a Global Navigation Satellite System (GNSS) sensor, including GPS functionality, and a photodiode sensor for photople-thysmography measurements. This device measures running biomechanics variables such as power, speed, cadence, vertical oscillation, and ground contact time. Participants also wore a GARMINRP HRM-PRO heart rate monitor below the pectoral zone in a centered and vertically oriented position. Power was assessed by the GARMINRP HRM-PRO heart rate monitor and speed, cadence, vertical oscillation, and ground contact time were measured by the GARMINRP Fenix 7S Solar watch”. On the other side, Stryd “device is a carbon fiber–reinforced foot pod based on a 6-axis inertial motion sensor (3-axis gyroscope and 3-axis accelerometer)”.

These variables are relevant for trail running performance as they provide insights into the runner's biomechanics and can help identify areas for improvement in technique and efficiency. For example, power output is a key determinant of running performance, and measuring power can help runners optimize their training and pacing strategies.

The specification “measured variables” has been added to report the variables obtained with each device, with the sentence (Lines 161-163): “Variables measured with this StrydTM technology include power, speed, cadence, vertical oscillation, and the duration of contact time.” Thank you for this comment, a result of a thorough and thoughtful review.

  1. To improve the clarity of the conclusion and discussion, it could be revised to provide a more concise summary of the study's key findings and their practical implications for trail runners and fitness enthusiasts. The conclusion could also be strengthened by providing specific recommendations for trail runners based on the study's results, such as guidelines for selecting and using wearable devices to monitor running biomechanics. Additionally, the conclusion could suggest areas for future research to further enhance the understanding of wearable devices' reliability in assessing running metrics.

ANSWER 2: Thank you for your feedback. We have taken your suggestions into consideration and have added the following paragraph to the discussion section (Lines 492-506): “Lastly, wearable technology, as evidenced by some previous evidence (27–30, hold promise in predicting and preventing injuries, enhancing sports biomechanics, and addressing musculoskeletal concerns, emphasizing the need for robust methodologies and clear reporting in research. In addition, real-time monitoring of biomechanical variables in mountain trail running emerges as a promising future perspective. This innovation would provide runners with the ability to receive instant feedback on their technique, enabling personalized adjustments during the run. Furthermore, early identification of inefficient biomechanical patterns could contribute to injury prevention, while optimizing performance on changing terrains would be possible through immediate adjustments. Advances in technologies such as wearable sensors and cloud-based data analysis would facilitate the implementation of this real-time monitoring, not only enhancing individual performance but also contributing to scientific research and the development of innovative strategies in the field of mountain trail running (28,29). Moreover, integrating mechanical, vertical, and metabolic data into apps and smartwatches enhances real-time insights for trail runners, allowing immediate adjustments and a more personalized, data-driven approach to optimize performance.”

We believe that this paragraph adds valuable insights into the potential future applications of wearable technology in mountain trail running, including real-time monitoring of biomechanical variables and the integration of mechanical, vertical, and metabolic data into apps and smartwatches. These innovations have the potential to enhance individual performance, prevent injuries, and contribute to scientific research and the development of innovative strategies in the field of trail running.

  1. To improve the reliability assessment of wearable devices, discuss and justify chosen methodologies. Consider alternative approaches and additional validation measures, such as comparing with gold standards or conducting extra statistical analyses, to strengthen reliability and enhance overall study validity.

ANSWER 3: Thank you for your valuable feedback. We have addressed the concerns regarding the reliability assessment of wearable devices by adding the following paragraph to the discussion section (Lines 508-513): “Nevertheless, measuring biomechanical variables in trail running faces challenges due to the absence of a universal gold standard. Wearable devices like inertial sensors and GPS trackers offer insights, but the lack of standardized approaches raises con-cerns about data reliability. Further research is crucial to refine methodologies, address individual biomechanical variations, and enhance the validity of measurements in trail running using existing wearable technologies.”

This paragraph acknowledges the challenges in establishing a universal gold standard for measuring biomechanical variables in trail running and emphasizes the need for further research to refine methodologies and enhance the validity of measurements using wearable technologies. We believe that this addition strengthens the discussion by highlighting the ongoing challenges and the importance of refining methodologies to ensure the reliability of wearable devices in assessing running metrics.

Reviewer 4 Report

Comments and Suggestions for Authors

Dear Authors,

Thank you very much for your interesting manuscript. The general topic you address is of high interest to the community of sport science and exercise physiology.

The abstract is well-written and concise. Your methods are appropriate for answering your research questions and are described adequately. Moreover, your presentation of results is adequate and includes appropriate figures and tables.

However, you work still requires substantial revision before I can recommend its publication in Sensors. In the following you will find a list of issues to be thoroughly addressed.

Major issues in general:

  • You do not provide any validity testing of the devices for use in trail running scenarios. The relevance of your findings for researchers and practitioners is severely limited if the devices are reliable but not valid.).
  • Your very small sample size of only 5 runners (including only 1 female) severely limits external validity.

Minor issues in general:

  • Language is at times imprecise and unnecessarily long-winded with a substantial amount of typos.
  • The key word "Trail" should be included in the title ('Assessing Trail Running Biomechanics: A Comparative Analysis of the Reliability of StrydTM and GARMINRP Wearable Devices')

Detailed feedback:

Introduction:

  • Long-winded and difficult-to-understand sentences (especially lines 30-39)
  • Does not demonstrate relevance in a clear and precise manner, e.g.: Is validity of spatio-temporal parameters important or is reliability sufficient for consumers? The first is true.
  • Contents redundancy in terms of unnecessary repetitions
  • Line of argumentation is difficult to follow.

Methods:

  • Language could be improved, some typos

Results:

  • Well written!

Discussion:

  • Language needs improvement (especially paragraph 285-316)
  • A major portion of your discussion focuses on the general validity of the Stryd sensor, resembling in parts a literature review, although the specific validity of the Stryd sensor for trail running would be of much higher interest to the reader at this point.
  • I recommend adding a paragraph highlighting the practical implications of your study's findings as the actual essence of your work.

Best regards

Your Reviewer

Comments on the Quality of English Language

English language needs moderate editing.

Author Response

REVIEWER 4

Dear Authors,

Thank you very much for your interesting manuscript. The general topic you address is of high interest to the community of sport science and exercise physiology.

The abstract is well-written and concise. Your methods are appropriate for answering your research questions and are described adequately. Moreover, your presentation of results is adequate and includes appropriate figures and tables.

However, you work still requires substantial revision before I can recommend its publication in Sensors. In the following you will find a list of issues to be thoroughly addressed.

Major issues in general:

  • You do not provide any validity testing of the devices for use in trail running scenarios. The relevance of your findings for researchers and practitioners is severely limited if the devices are reliable but not valid.).
    • Answer: Thank you for your comment. Thank you for your insightful comment. We acknowledge the importance of validity testing for wearable devices in trail running scenarios. While our study focused on assessing the reliability of the devices, we recognize that validity testing is crucial to fully understand the applicability of the findings for researchers and practitioners. However, the absence of a universal gold standard for measuring biomechanics in trail running in natural environments is a significant limitation and challenge, as mentioned in the manuscript (lines 508-511): “Nevertheless, measuring biomechanical variables in trail running faces challenges due to the absence of a universal gold standard. Wearable devices like inertial sensors and GPS trackers offer insights, but the lack of standardized approaches raises con-cerns about data reliability” . This makes it difficult to establish the validity of wearable devices and other measurement tools.Therefore, in our study, we focused on assessing the reliability of wearable devices in measuring running biomechanics in trail running scenarios. While we acknowledge the importance of validity testing, the lack of a gold standard for measuring biomechanics in trail running in natural environments makes it challenging to establish the validity of the devices.
  • Your very small sample size of only 5 runners (including only 1 female) severely limits external validity.
    • Answer: Thank you for providing valuable feedback. We value your perspectives and have thoroughly taken into account the recommendations you shared. Concerning the sample size, we recognize the significance of statistical power, especially for 0-dimensional data like power, speed, and cadence. Although the participant count might seem limited, it's essential to highlight that our primary emphasis is on intra and inter-device agreement analysis, generating a substantial volume of data points even with a restricted participant pool. Nevertheless, we acknowledge the methodological limitation in this aspect.

Minor issues in general:

  • Language is at times imprecise and unnecessarily long-winded with a substantial amount of typos.
    • Answer: Thank you for your feedback. We have carefully reviewed and revised the manuscript to ensure that the language is clear, concise, and free of errors.
  • The key word "Trail" should be included in the title ('Assessing Trail Running Biomechanics: A Comparative Analysis of the Reliability of StrydTM and GARMINRP Wearable Devices')
    • Answer: Thank you for your suggestion. We have incorporated the key term 'Trail' into the title, and it now reads as follows: “Assessing Trail Running Biomechanics: A Comparative Analysis of the Reliability of StrydTM and GARMINRP Wearable Devices” We appreciate your attention to detail and input on enhancing the clarity of our work.

Detailed feedback:

Introduction:

  • Long-winded and difficult-to-understand sentences (especially lines 30-39)
    • Answer: Thank you for highlighting your concerns. We have revisited the specified lines (30-39) and addressed the issue of long-winded and difficult-to-understand sentences. The content has been revised for improved clarity and conciseness. We appreciate your feedback, which has improved the manuscript.
  • Does not demonstrate relevance in a clear and precise manner, e.g.: Is validity of spatio-temporal parameters important or is reliability sufficient for consumers? The first is true.
    • Answer:  We appreciate your thoughtful comments and welcome the opportunity to address your concerns regarding the relevance of validity and reliability in our trail running biomechanics research.

Regarding the validity of spatio-temporal parameters, we acknowledge the significance of this dimension in assessing running biomechanics. However, it is crucial to emphasize that, as of the date of our research, there is no gold standard for biomechanical measurements in natural or trail running conditions due to technical and logistical limitations in this specific context.  

Consequently, while we recognize the importance of validity and share the interest in its consideration, the current reality leads us to focus on the reliability of measurements. Reliability provides a critical assessment of the consistency and reproducibility of results obtained with devices such as GARMIN and Stryd in real trail running environments.  

In our study, we have demonstrated high intra-device reliability and consistency in measurements from both GARMIN and both Styd devices across various performance variables. This emphasis on reliability is essential to ensure that the measurements made by these devices are consistent and reproducible in trail running settings, which is crucial for their practical applicability.  

We appreciate the suggestion regarding the importance of validity and acknowledge its relevance in the biomechanical field. However, the lack of a gold standard in natural settings leads us to focus on reliability as a crucial indicator for the practical utility of these devices in trail running.

  • Contents redundancy in terms of unnecessary repetitions
    • Answer: Thank you for your feedback regarding the redundancy in the contents of the manuscript. We appreciate your insight and we have carefully reviewed the document to identify and eliminate any unnecessary repetitions. Our goal is to ensure that the manuscript is clear, concise, and free of redundant information.
  • Line of argumentation is difficult to follow.
    • Answer: Thank you for your feedback regarding the line of argumentation in the introduction of our manuscript. We understand that it is important to present a clear and logical argument to guide the reader through the content of the paper.

In our introduction, we aimed to provide a comprehensive overview of the complexity of trail running as a sport, highlighting the numerous performance factors involved, including aerobic capacity, anaerobic threshold, muscular strength, and resistance to force due to both positive and negative elevation changes. We also emphasized the importance of biomechanical adaptations in uphill and downhill running, including changes in foot strike pattern, joint kinematics, and energy cost.

We believe that this approach provides a clear and logical argument for the need to investigate the reliability of wearable devices in measuring spatio-temporal parameters in trail running scenarios. By establishing the complexity of the sport and the importance of biomechanical adaptations, we set the stage for the relevance of wearable devices in measuring these parameters.

In addition, we believe that the current structure of the introduction provides a strong foundation for contextualizing the importance of assessing the reliability of wearable devices in the context of trail running. However, we have already taken steps to improve the clarity of the argument in the introduction by restructuring some paragraphs. We understand the importance of presenting a clear and logical argument, and we are committed to ensuring that the manuscript is easy to follow and understand. Lastly, we are open to discussing any proposals for restructuring that may improve the coherence and clarity of the argument in the introduction. We value your feedback and are willing to consider any suggestions that may strengthen the presentation of the argument in the paper.

Methods:

  • Language could be improved, some typos
    • Answer: Thank you for your feedback. We have carefully reviewed the document and addressed the issues you pointed out. Typos have been corrected, and we have tried to improve the language to ensure greater clarity and fluency. We appreciate your attention to detail, and we are committed to delivering high-quality content.

Results:

  • Well written!
    • Answer: Thank you for your positive feedback! We are delighted to hear that you found the content well-written.

Discussion:

  • Language needs improvement (especially paragraph 285-316)
    • Answer: We appreciate the reviewer's feedback regarding the language in the specified paragraphs. We acknowledge the importance of clear and effective language in scientific writing and are committed to addressing this concern. Therefore, we have carefully reviewed the identified paragraphs to improve the language and ensure that the content is presented in a clear and coherent manner. Additionally, we have also considered revising the language to enhance readability and comprehension.
  • A major portion of your discussion focuses on the general validity of the Stryd sensor, resembling in parts a literature review, although the specific validity of the Stryd sensor for trail running would be of much higher interest to the reader at this point.
    • Answer: Thank you for your feedback. We've revised the discussion to focus specifically on the validity of the Stryd sensor for trail running, addressing observed discrepancies with GARMINRP. As mentioned in other comments, the absence of a universal gold standard for outdoor measurements is acknowledged as a shared challenge in the field. We appreciate your insights and welcome any additional suggestions.
  • I recommend adding a paragraph highlighting the practical implications of your study's findings as the actual essence of your work.
    • Answer: Thank you for your recommendation. In response to your suggestion, we have added a dedicated paragraph in the discussion section to highlight the practical implications of our study's findings (lines 480-522):

“It is worth noting the potential impact of terrain and environmental conditions on the accuracy of wearable devices, as demonstrated in a recent article by and Uwe Schlink (26). This article sheds light on the challenges and applications of wearable sensors in diverse environmental conditions and terrains, emphasizing the influence of recording intervals on sensor performance. The authors underscore the necessity of investigating the accuracy of running metrics measurements obtained with wearable devices in different terrains and environmental conditions. This can be achieved by testing the devices in specific contexts of trail running sports modalities to evaluate the intra-device reliability of the wearable devices. The results obtained in the present re-search reveal discrepancies between measurements obtained with StrydTM and GARMINRP. These observed variations may be attributed to inherent differences in the algorithms, sensor technologies, or calibration methods employed by both devices.

Lastly, wearable technology, as evidenced by some previous evidence (27–30, hold promise in predicting and preventing injuries, enhancing sports biomechanics, and addressing musculoskeletal concerns, emphasizing the need for robust methodologies and clear reporting in research. In addition, real-time monitoring of biomechanical variables in trail running emerges as a promising future perspective. This innovation would provide runners with the ability to receive instant feedback on their technique, enabling personalized adjustments during the run. Furthermore, early identification of inefficient biomechanical patterns could contribute to injury prevention, while optimizing performance on changing terrains would be possible through immediate adjustments. Advances in technologies such as wearable sensors and cloud-based data analysis would facilitate the implementation of this real-time monitoring, not only enhancing individual performance but also contributing to scientific research and the development of innovative strategies in the field of trail running (28,29). Moreover, integrating mechanical, vertical, and metabolic data into apps and smartwatches enhances real-time insights for trail runners, allowing immediate adjustments and a more personalized, data-driven approach to optimize performance.

Nevertheless, measuring biomechanical variables in trail running faces challenges due to the absence of a universal gold standard. Wearable devices like inertial sensors and GPS trackers offer insights, but the lack of standardized approaches raises concerns about data reliability. Further research is crucial to refine methodologies, address individual biomechanical variations, and enhance the validity of measurements in trail running using existing wearable technologies.

Our research approach stands out for its innovative stride in addressing the existing gap in the biomechanical evaluation of trail running under real-world conditions. While past studies have primarily relied on laboratory settings, our research takes a pioneering step by harnessing the power of recent technological advancements in wearable devices. By exploring the reliability and consistency of GARMINRP and StrydTM devices in measuring key biomechanical variables during trail running, including analyses for both uphill and downhill scenarios, our research provides a novel contribution to the field, offering valuable insights with practical implications for both re-searchers and practitioners in trail running and sports science.”.

The paragraph emphasizes the potential impact of terrain and environmental conditions on the accuracy of wearable devices, as supported by recent research (26). It also discusses the promise of wearable technology in predicting and preventing injuries, enhancing sports biomechanics, and addressing musculoskeletal concerns. Additionally, the paragraph explores the potential of real-time monitoring of biomechanical variables in trail running, emphasizing its benefits in injury prevention and performance optimization. We believe that this addition enhances the overall practical relevance of our study's outcomes.

Round 2

Reviewer 1 Report

Comments and Suggestions for Authors

The authors addressed my concerns.

Author Response

We sincerely appreciate your thoughtful review of our work. We are pleased to hear that the revisions we made have effectively addressed your concerns. Your feedback has been invaluable in enhancing the quality and clarity of our manuscript. Thank you for your time and insightful comments, which have undoubtedly strengthened the overall contribution of our research.

Reviewer 2 Report

Comments and Suggestions for Authors

I appreciate the revision made by the authors, which shows a significant improvement to the manuscript quality. I also appreciate the hard work by the authors during the manuscript writing and revision. However, I still consider there exist some flaws in the study designs. While the authors have convinced me regarding the intra-device comparison in this round, the inter-device comparison still confuses me as its rationale is not clear. Due to the lack of comparison against a gold standard, it is difficult to tell which device can do a better job in biomechanics measurement. As stated in the Discussion section, previous studies reported significant differences between the Garmin and Stryd wearable devices against the gold standard for some variables, I am wondering the rationale for comparing these two devices which are ‘not accurate’. After detecting differences between the two, we cannot tell which one is accurate or more accurate. It looks like ‘the two devices are different’ could be the only main finding for this part of this study. Based on such finding, I am unable to ‘provide valuable information for both researchers and practitioners’. Yes, this study pioneered the measurement of trail running in real-world scenario using wearable devices, but the authors are recommended to re-think the study design.

Regarding the writing of this manuscript, besides some grammar issues, some parts should be re-organised to provide a better flow regardless of the study design. The Discussion is very lengthy, but most parts are not discussing surrounding the current research results. This makes it look like Introduction rather than Discussion. However, some parts (such as the reason for intra-device analysis) should be relocated under Introduction.

Below are some specific comments based on the revision and responses from the authors. In addition, I guess some of the changes are only shown on the ‘cover letter’ but not on the revised manuscript (possibly due to the use of the ‘Track Changes’ function). The authors should check again to provide a corrected version.

1.     Regarding the term ‘power’ used in this study, I did not request the algorithm behind it. A clear definition is necessary. In the biomechanics context, power can refer to joint power (joint kinetics, e.g., ankle joint power, knee joint power). However, in this study, power seems to relate to energy expenditure for the whole body while running, if I am not wrong. This is the reason this term should be clarified. This variable is usually measured in physiology studies, but it may not be wrong to consider it as a biomechanics variable.

2.     In Abstract, it is necessary to add some detailed results such as the comparison between devices as the key findings of this study.

3.     I agree with the responses provided by the authors regarding the need to conduct intra-device reliability analysis. The points are good. Please also consider including those lines in Introduction (now under Discussion), since convincing readers who can only read the final version is more important. However, as mentioned by the authors on the ‘cover letter’, different device conditions can influence the measurement accuracy. Did the authors purposely used devices in different conditions (e.g., new versus old) to conduct the intra-device analysis? In the study, was the good agreement in the intra-device analysis due to the devices being in similar conditions (e.g., 2 new sets)? I feel the paragraph starting in Line 100 may not be truly relevant to the main topic of this study. Yes, treadmill running and overground running are different as pointed out by previous researchers. One or 2 lines on this can be included in the previous paragraph (just after ‘laboratory settings’). It is not necessary to have a stand-alone paragraph to talk about this point not very close to the main topic of this study.

4.     Line 107. This paragraph can be more concise, while it now looks to have so many research purposes. Actually, there are only 2 purposes (inter- and intra-device reliability analyses). A concise statement helps readers follow the contents easily and allows them to memorise the main topics easily.

5.     Line 176. Please consider adding a line to briefly state that a low sampling frequency is usually employed for measurements using wearable devices. It will be better to also include the references mentioned on the ‘cover letter’.

6.     Line 191. This is a minor issue, but a clearer statement will be helpful. For the intra-device comparison, it is not clear which trials were used (e.g., comparison between the first and second trials). It is also unclear for the inter-device comparison (e.g., comparison between the first trials of the 2 brands).

Comments on the Quality of English Language

I guess some of the changes are only shown on the ‘cover letter’ but not on the revised manuscript (possibly due to the use of the ‘Track Changes’ function). The authors should check again to provide a corrected version.

Author Response

I appreciate the revision made by the authors, which shows a significant improvement to the manuscript quality. I also appreciate the hard work by the authors during the manuscript writing and revision. However, I still consider there exist some flaws in the study designs. While the authors have convinced me regarding the intra-device comparison in this round, the inter-device comparison still confuses me as its rationale is not clear. Due to the lack of comparison against a gold standard, it is difficult to tell which device can do a better job in biomechanics measurement. As stated in the Discussion section, previous studies reported significant differences between the Garmin and Stryd wearable devices against the gold standard for some variables, I am wondering the rationale for comparing these two devices which are ‘not accurate’. After detecting differences between the two, we cannot tell which one is accurate or more accurate. It looks like ‘the two devices are different’ could be the only main finding for this part of this study. Based on such finding, I am unable to ‘provide valuable information for both researchers and practitioners’. Yes, this study pioneered the measurement of trail running in real-world scenario using wearable devices, but the authors are recommended to re-think the study design.

Regarding the writing of this manuscript, besides some grammar issues, some parts should be re-organised to provide a better flow regardless of the study design. The Discussion is very lengthy, but most parts are not discussing surrounding the current research results. This makes it look like Introduction rather than Discussion. However, some parts (such as the reason for intra-device analysis) should be relocated under Introduction.

Below are some specific comments based on the revision and responses from the authors. In addition, I guess some of the changes are only shown on the ‘cover letter’ but not on the revised manuscript (possibly due to the use of the ‘Track Changes’ function). The authors should check again to provide a corrected version.

ANSWER: We appreciate your time spent reviewing our manuscript and for acknowledging the improvements made in response to your feedback. We highly value your comments and the opportunity to address them.

Thank you for recognizing the hard work invested in both the writing and revision processes. We understand your concerns regarding the study design, particularly regarding the inter-device comparison. While we aimed to provide insights into the performance of wearable devices in real-world trail running scenarios.

Regarding the organization of the manuscript, we wish to maintain the structure of the Introduction and Discussion sections as they currently stand. We believe that they follow a coherent narrative that enhances understanding. We have addressed the grammar issues. Additionally, we have carefully reviewed the changes made in response to your comments to ensure they are accurately reflected in the revised manuscript. Thank you again for your constructive feedback.

  1. Regarding the term ‘power’ used in this study, I did not request the algorithm behind it. A clear definition is necessary. In the biomechanics context, power can refer to joint power (joint kinetics, e.g., ankle joint power, knee joint power). However, in this study, power seems to relate to energy expenditure for the whole body while running, if I am not wrong. This is the reason this term should be clarified. This variable is usually measured in physiology studies, but it may not be wrong to consider it as a biomechanics variable.

ANSWER: Thank you for your comment regarding the term 'power' used in our study. We acknowledge the importance of providing a clear definition for this variable, especially considering its relevance within the context of biomechanics. In response to your feedback, we have added this paragraph in the manuscript clarifying the definition of power.

“The measured power specifically refers to external mechanical power, calculated from force (estimated via accelerations using accelerometers) and velocity. This includes the work exerted by runners during both the loading phase and subsequent push-off to counteract environmental factors such as ground reaction force, gravity, and surface friction” (lines 152-156).

We hope that this clarification addresses your concerns and enhances the understanding of our study findings.

  1. In Abstract, it is necessary to add some detailed results such as the comparison between devices as the key findings of this study.

ANSWER: Thank you for your comment regarding the abstract of our study. We appreciate your attention to detail and agree that providing detailed results, especially regarding the comparison between devices, is essential for clarity. We believe that the statement included in the abstract, "distinctions emerged in inter-device agreement, particularly in power and contact time uphill, and vertical oscillation downhill, suggesting potential variations between GARMINRP and StrydTM measurements for specific running metrics," adequately addresses this aspect. We aimed to highlight key findings related to device comparison within the limited space of the abstract, and we believe that this statement effectively conveys the essential results of our study in this regard.

  1. I agree with the responses provided by the authors regarding the need to conduct intra-device reliability analysis. The points are good. Please also consider including those lines in Introduction (now under Discussion), since convincing readers who can only read the final version is more important. However, as mentioned by the authors on the ‘cover letter’, different device conditions can influence the measurement accuracy. Did the authors purposely used devices in different conditions (e.g., new versus old) to conduct the intra-device analysis? In the study, was the good agreement in the intra-device analysis due to the devices being in similar conditions (e.g., 2 new sets)? I feel the paragraph starting in Line 100 may not be truly relevant to the main topic of this study. Yes, treadmill running and overground running are different as pointed out by previous researchers. One or 2 lines on this can be included in the previous paragraph (just after ‘laboratory settings’). It is not necessary to have a stand-alone paragraph to talk about this point not very close to the main topic of this study.

ANSWER: Thank you for your thorough feedback and suggestions regarding our manuscript. We appreciate your attention to detail and the opportunity to address your concerns.

Regarding the intra-device reliability analysis, we confirm that the devices used in our study were all acquired simultaneously and were of the same version. While there may be standardization in manufacturing, each device is equipped with its own hardware specific to the respective watch model, which could potentially lead to discrepancies. This clarification aims to provide insight into the consistency of conditions across devices in our study.

We have also taken your suggestion into consideration regarding the organization of the text for clarity. The paragraph that you mentioned has been included in the previous one, and the new paragraph can be read as follows: “The comparison between motorized treadmill running and overground running highlights variations in sagittal plane measures and spatiotemporal parameters, with conflicting findings across analyses of kinematics, kinetics, muscle activity, and muscle-tendon outcomes (15). Notably, the comparison does not exclusively involve trail running, which presents unique characteristics in overground locomotion (15).” (lines 84-88).

Furthermore, we have addressed the term "accuracy" to avoid confusion, replacing it with "agreement" where necessary.

We hope that these revisions address your concerns and contribute to the overall clarity and coherence of the manuscript. Thank you for your valuable input, which has helped enhance the quality of our work.

  1. Line 107. This paragraph can be more concise, while it now looks to have so many research purposes. Actually, there are only 2 purposes (inter- and intra-device reliability analyses). A concise statement helps readers follow the contents easily and allows them to memorise the main topics easily.

ANSWER: Thank you for your feedback. We have revised the paragraph to make it more concise and focused. The updated version now reads as follows: “This study aims to investigate and compare the biomechanical data reported by the two most popular wearable devices, GARMINRP and StrydTM, which provide measures of biomechanics during trail running under natural conditions. Specifically, our objectives include, on the one hand, assessing the consistency and agreement of biomechanical measurements within the GARMINRP or StrydTM devices by comparing one device against another. On the other hand, we seek to investigate the inter-device reliability and agreement by comparing the biomechanical parameters measured by a StrydTM device against a GARMINRP device. By conducting these comparisons, we aim to provide insights into the reliability and consistency of these wearable devices in measuring key biomechanical variables, offering valuable information for both researchers and practitioners in the field of trail running and sports science.” (Lines 100-119). We believe that these revisions improve clarity and streamline the paragraph, making it easier for readers to understand the main purposes of the study. Thank you for your input.

  1. Line 176. Please consider adding a line to briefly state that a low sampling frequency is usually employed for measurements using wearable devices. It will be better to also include the references mentioned on the ‘cover letter’.

ANSWER: Thank you for your valuable suggestion regarding Line 176. We have taken your feedback into consideration and have added a line to briefly mention that a low sampling frequency is typically employed for measurements using wearable devices. Additionally, we have included references to support this statement, as mentioned in the cover letter. The paragraph now reads as follows: "a sampling frequency similar to that used in similar studies on running biomechanics (5,6,14)." (Line 181-183). We believe that these additions enhance the clarity and comprehensiveness of the manuscript.

  1. Line 191. This is a minor issue, but a clearer statement will be helpful. For the intra-device comparison, it is not clear which trials were used (e.g., comparison between the first and second trials). It is also unclear for the inter-device comparison (e.g., comparison between the first trials of the 2 brands).

ANSWER: Thank you for your attention to detail and feedback regarding Line 191. We have addressed your concerns by adding a clearer statement regarding the trials used for both the intra-device and inter-device comparisons: “For intra-device comparison, measurements from one GARMINRP device were com-pared to a second GARMINRP device, both worn by the same individual on the same wrist. Similarly, measurements from one StrydTM device were compared to a second StrydTM device, both worn by the same individual on the foot. As stated before, the tests were conducted on different days following this comparison protocol” (Lines 208-213). We believe that these additions provide clarity regarding the methodology employed for the comparisons and enhance the transparency of our study.

Reviewer 4 Report

Comments and Suggestions for Authors

Dear Authors,

Thank you for providing a revised version of your manuscript.

I understand that there are general methodological limitations of your study that you cannot properly address without repeating the entire measurement procedure, especially regarding the still very low number of subjects and the missing, but highly important, evaluation of validity, instead of only addressing reliability.

Therefore, even though I still have concerns about the overall significance and the true practical scientific applicability of your study’s results, I acknowledge that you have made substantial effort to justify the publication of your work in Sensors.

In order to at least minimally address the lack of validation, I recommend that you compare your spatio-temporal gait results with recent speed-resolved values from the literature. Lang et al. (https://doi.org/10.3390/sports11100204, MDPI Sports) for instance, have recently published a comprehensive dataset on such parameters in junior elite runners, which may represent a sound reference even though it concerns primarily track/road racing and not cross country. You may be able to find a second additional reference to discuss in combination with this, especially regarding possible differences between trail and road running.

Moreover, please double-check that your grammar changes have been adapted correctly. In the current version, I see words with identical meanings (especially verbs, red and black in colour) directly following each other, which, of course, is grammatically incorrect. The first example of this error can be found in line 1, third and fourth word of the abstract (“The study explores investigates biomechanical assessments…”).

Best regards

Yours reviewer

Comments on the Quality of English Language

Doube-check your grammar changes.

Author Response

Dear Authors,

Thank you for providing a revised version of your manuscript.

I understand that there are general methodological limitations of your study that you cannot properly address without repeating the entire measurement procedure, especially regarding the still very low number of subjects and the missing, but highly important, evaluation of validity, instead of only addressing reliability.

Therefore, even though I still have concerns about the overall significance and the true practical scientific applicability of your study’s results, I acknowledge that you have made substantial effort to justify the publication of your work in Sensors.

In order to at least minimally address the lack of validation, I recommend that you compare your spatio-temporal gait results with recent speed-resolved values from the literature. Lang et al. (https://doi.org/10.3390/sports11100204, MDPI Sports) for instance, have recently published a comprehensive dataset on such parameters in junior elite runners, which may represent a sound reference even though it concerns primarily track/road racing and not cross country. You may be able to find a second additional reference to discuss in combination with this, especially regarding possible differences between trail and road running.

ANSWER: Thank you for your thoughtful consideration of our revised manuscript and for your understanding of the methodological limitations inherent in our study. We appreciate your acknowledgment of the efforts we've made to address these limitations.

Regarding your recommendation to address the lack of validation, we have incorporated a paragraph in the discussion section where we compare our spatio-temporal gait results with the article you suggested, as we have not found similar articles on trial running, which could be a topic for future research. The added paragraph can be read as follows: “However, although biomechanical variables seem to differentiate depending on whether running is performed on a treadmill or in real trail running conditions, these variables appear to behave similarly based on speed, as Lang et al. (26) showed in the results of their article. Despite higher cadence and shorter ground contact times on the treadmill compared to trail running, cadence increases downhill with speed and ground contact times decrease similarly in both settings (26). However, vertical oscilla-tion diverges; while it decreases with speed in laboratory tests, it increases on negative slopes in trail running where speed is higher (26). This discrepancy is likely attributed to terrain slope, which significantly impacts biomechanical and physiological adapta-tion, alongside adjustments in center of gravity during uphill and downhill.” Lines 491-500. We believe that this addition enhances the context and relevance of our study findings within the existing literature. Thank you again for your feedback and suggestions.

Moreover, please double-check that your grammar changes have been adapted correctly. In the current version, I see words with identical meanings (especially verbs, red and black in colour) directly following each other, which, of course, is grammatically incorrect. The first example of this error can be found in line 1, third and fourth word of the abstract (“The study explores investigates biomechanical assessments…”).

ANSWER: Thank you for bringing this to our attention. We have thoroughly reviewed our manuscript to ensure that all grammar changes have been correctly adapted. we have addressed this issue, including the specific example you provided in line 1 of the abstract. We appreciate your diligence in identifying these errors, and we are committed to ensuring the accuracy and clarity of our manuscript.
